# Molecular basis of ALK1-mediated signalling by BMP9/BMP10 and their prodomain-bound forms

Richard M. Salmon [1,4], Jingxu Guo[1,4], Jennifer H. Wood[1], Zhen Tong [1], John S. Beech[2], Aleksandra Lawera[1], Minmin Yu[3], David J. Grainger[2], Jill Reckless[2], Nicholas W. Morrell [1] & Wei Li [1✉]

Activin receptor-like kinase 1 (ALK1)-mediated endothelial cell signalling in response to bone morphogenetic protein 9 (BMP9) and BMP10 is of significant importance in cardiovascular disease and cancer. However, detailed molecular mechanisms of ALK1-mediated signalling remain unclear. Here, we report crystal structures of the BMP10:ALK1 complex at 2.3 Å and the prodomain-bound BMP9:ALK1 complex at 3.3 Å. Structural analyses reveal a tripartite recognition mechanism that defines BMP9 and BMP10 specificity for ALK1, and predict that crossveinless 2 is not an inhibitor of BMP9, which is confirmed by experimental evidence. Introduction of BMP10-specific residues into BMP9 yields BMP10-like ligands with diminished signalling activity in C2C12 cells, validating the tripartite mechanism. The loss of osteogenic signalling in C2C12 does not translate into non-osteogenic activity in vivo and BMP10 also induces bone-formation. Collectively, these data provide insight into ALK1-mediated BMP9 and BMP10 signalling, facilitating therapeutic targeting of this important pathway.

---

[1] The Department of Medicine, University of Cambridge School of Clinical Medicine, Cambridge CB2 0QQ, UK. [2] RxCelerate Ltd, Babraham Research Campus, Cambridge CB22 3AT, UK. [3] MRC Laboratory of Molecular Biology, Francis Crick Avenue, Cambridge CB2 0QH, UK. [4] These authors contributed equally: Richard M. Salmon, Jingxu Guo. ✉email: wl225@cam.ac.uk

Bone morphogenetic proteins (BMPs) are dimeric transforming growth factor β (TGFβ) family cytokines, initiating cellular signalling by forming a complex with two copies of type I receptors and two copies of type II receptors, both of which are type I transmembrane proteins with an intracellular serine/threonine kinase domain. Activin receptor-like kinase 1 (ALK1) is a type I receptor specifically expressed on endothelial cells[1], mediating the signals from BMP9 and BMP10 exclusively[2]. BMP signalling can be regulated by co-receptors, such as endoglin (ENG), as well as extracellular ligand traps, such as noggin and crossveinless 2 (CV2)[3]. All TGFβ family ligands are synthesised as preproproteins, processed by furin-like proprotein convertase upon secretion. It has been shown for several family members, including TGFβ, BMP9 and BMP10, that the prodomain remains tightly bound to the growth factor (GF) domain in circulation[4–6]. Hence, the prodomain may also contribute to the regulation of BMP signalling.

BMP9 and BMP10 share a high degree of sequence identity (64% in the GF-domain and 33% in the prodomain); both ligands induce similar target genes in endothelial cells[7] and BMP10 is able to compensate for the function of BMP9 in *Bmp9* knockout mice[7,8]. Compared with other BMPs, BMP9 and BMP10 possess several unique features. Firstly, both ligands bind to ALK1 and ENG with sub-nanomolar affinities[9,10]; and the affinities of both ligands for their cognate type I receptor ALK1 ($EC_{50}$ of around 50 pg ml$^{-1}$) are much higher than those of other BMPs for their cognate type I receptors ALK2, ALK3, and ALK6 ($EC_{50}$ of around 50 ng ml$^{-1}$)[4]. Secondly, whilst several naturally-occurring extracellular ligand traps have been described for BMPs, most of them do not inhibit BMP9 or BMP10 signalling[3], apart from reports that CV2 binds to BMP9 and BMP10 and suppresses their signalling[11,12].

ALK1-mediated endothelial BMP9 and BMP10 signalling plays many important roles in angiogenesis and the maintenance of vascular quiescence[13,14]. Defects in this pathway are known to cause cardiovascular diseases. For instance, individuals with heterozygous loss-of-function mutations in *ALK1* or *ENG* develop hereditary haemorrhagic telangiectasia (HHT)[15,16], and *BMP9* mutations have been identified in patients with a phenotype similar to HHT[17]. Typical HHT manifestations involve vascular abnormalities such as arteriovenous malformations in the brain, lung, liver, or gastrointestinal tract. In addition, heterozygous loss-of-function mutations in the type II BMP receptor *BMPR2*, as well as in *ALK1*, *ENG*, and *BMP9*, have been identified in patients with pulmonary arterial hypertension (PAH)[18–21]. The pathophysiology of PAH includes endothelial dysfunction and vascular remodelling, resulting in narrowing of pulmonary arteries, elevated pulmonary arterial pressure and right ventricular heart failure[22]. Importantly, targeting the endothelial BMP pathway has promising therapeutic potential. For example, administration of recombinant BMP9 prevented and reversed the disease and inhibited angiogenesis in preclinical PAH models[23]. Moreover, the ALK1 extracellular domain-Fc fusion protein (ALK1-Fc, Dalantercept) and an anti-ENG antibody (TRC105) have demonstrated anti-tumour angiogenesis activity and are currently in phase II clinical trials for treating selected solid tumours[24–26].

A comprehensive understanding of the molecular mechanisms behind BMP9 and BMP10 signalling will provide important information for translating therapies that target the endothelial BMP pathway. There are a number of questions that remain to be answered, particularly regarding the difference between BMP9 and BMP10 as well as the regulatory role of the prodomain. Although BMP9 and BMP10 have been shown to induce several identical genes in endothelial cells[7], no direct comparison of the global transcription regulated by these two ligands has been performed to date, especially by their circulating prodomain-bound forms (pro-BMP9 and pro-BMP10, see Supplementary Fig. 1). In addition, despite several crystal structures of BMP9 being solved, either as GF-domain alone or GF-domain in complex with receptors, the co-receptor or the prodomain[9,27–30], crystal structures of BMP10 remain to be elucidated.

In this study, we directly compare global gene expression regulated by the circulating forms of pro-BMP9 and pro-BMP10 using microarray. We report the crystal structures of BMP10 GF-domain in complex with ALK1 to 2.3 Å and prodomain-bound BMP9 in complex with ALK1 to 3.3 Å. Structural and sequence analyses alongside previously reported BMP9 structures[9,29,30] enable the identification of two conserved regions in BMP9 and BMP10 that define their specificity for ALK1, ENG and their prodomains. Furthermore, our structural analysis suggests that CV2 does not inhibit BMP9 signalling, which we confirm by experimental evidence, in contrast to a previous report[11]. Finally, guided by the structural analysis, we modify BMP9 signalling specificity by single amino acid substitutions. Surprisingly, we find that in vitro alkaline phosphatase (ALP)-based osteogenic signalling activity does not correlate with in vivo heterotopic ossification and that previously reported non-osteogenic BMP10 can induce bone formation in vivo.

## Results

**Pro-BMP9 and pro-BMP10 are equivalent ALK1 ligands.** Although pro-BMP9 and pro-BMP10 are the in vivo circulating forms of ligands[4,5], most of the previous signalling and functional studies on BMP9 and BMP10 have employed the GF-domain alone. Thus, it is essential to establish whether pro-BMP9 and pro-BMP10 bind to ALK1 and signal in endothelial cells in the same manner as their GF-domains. We first compared whether pro-BMP9 and pro-BMP10 can signal with similar potency to their GF-domains. As shown in Fig. 1a, in a dose-response assay in human pulmonary artery endothelial cells (PAECs) using *Inhibitor of DNA binding protein 1* (*ID1*) gene induction as a readout, pro-BMP9 and pro-BMP10 exhibited identical potency to BMP9 and BMP10 GF-domains, respectively.

We then carried out a microarray experiment to establish whether there is any differential gene induction by pro-BMP9 and pro-BMP10 in endothelial cells. As shown in Fig. 1b–d and Supplementary Data 1 and 2, both pro-BMP9 and pro-BMP10 induced and suppressed similar transcript expression when compared with PBS-treated controls; indeed, there was no significant difference in gene induction or suppression when comparing pro-BMP9 with pro-BMP10, providing evidence that they are functionally equivalent ALK1 ligands in vitro on endothelial cells. We further compared the ALK1 binding affinity of pro-BMP9 and pro-BMP10 to that of BMP9 and BMP10, respectively, using surface plasmon resonance (SPR). As shown in Fig. 1e, f, the $K_D$ values measured for BMP9:ALK1-Fc and BMP10:ALK1-Fc (48.1 pM and 34.3 pM, respectively) were very similar to those previously reported (45.22 pM and 10.3 pM, respectively)[9]. In addition, both BMP9 and BMP10 bound to the ALK1 monomer (produced in house)[28] and dimer (ALK1-Fc) with similar affinities, irrespective of the presence of their prodomains (Fig. 1e, f), which supported the findings from Fig. 1a that the prodomain does not interfere with ALK1-mediated signalling potency.

**Structure of BMP10:ALK1 complex.** To delineate the molecular details of ALK1-mediated BMP10 signalling, we solved the crystal structure of the BMP10 GF-domain in complex with the ALK1 extracellular domain (ECD) to 2.3 Å (Fig. 2a and Supplementary Table 1; Supplementary Figs. 2, 3). As expected, the ALK1 ECD

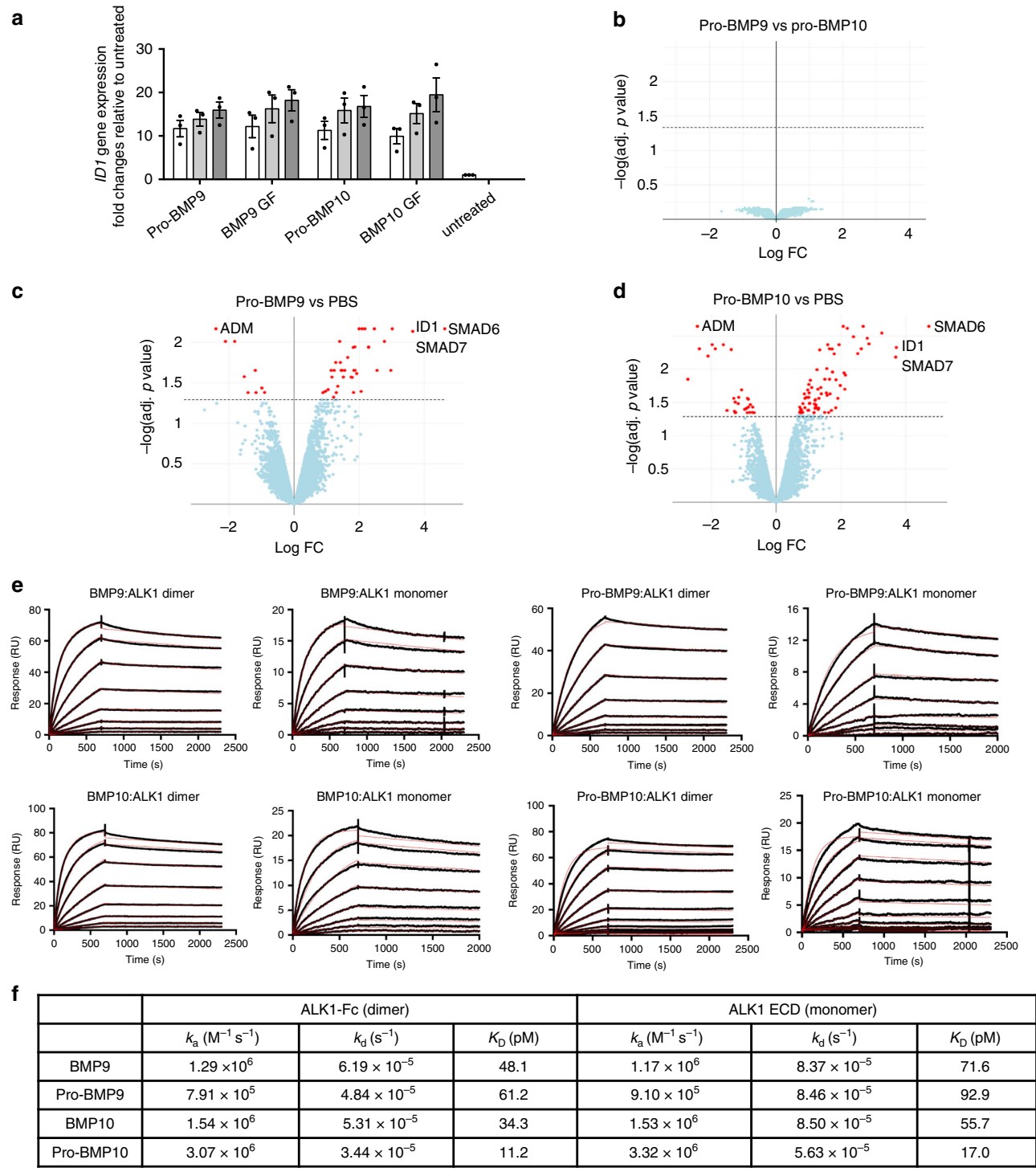

**Fig. 1 Pro-BMP9 and pro-BMP10 are equivalent ALK1-ligands. a** Dose-dependent signalling assays in PAECs. Serum-starved PAECs were treated with different ligands at 2.48 pM (white bars), 8.27 pM (light grey bars) and 27.3 pM (dark grey bars) (using monomer molecular weight, equivalent to 0.03, 0.1 and 0.33 ng ml$^{-1}$ BMP9 GF-domain concentration) for 1 h. Changes in the *ID1* gene expression were monitored using RT-qPCR. Data were presented as fold change relative to untreated cells, and means ± SEM of three independent experiments are shown. Source data are provided as a Source Data file. **b**–**d** Volcano plots comparing changes in global gene expression in PAECs after pro-BMP9 or pro-BMP10 treatment. Serum-starved PAECs were treated with 25 pM of pro-BMP9 or pro-BMP10 (purity can be found on SDS-PAGE with silver staining in Supplementary Fig. 8a, lanes 1 and 4) for 1.5 h before RNA was extracted for microarray analysis. Four different primary PAEC lines were used. Red dots above the dashed line represent the changes in target genes with adjusted *p* values of less than 0.05. Several representative target genes are highlighted in **c** and **d**. Full list of genes can be found in Supplementary Data 1 and 2. **e** Affinity measurements of BMP9 and BMP10 for ALK1 using Biacore. A CM5 Biacore chip was immobilised with ALK1 dimer (ALK1-Fc) or monomer (in-house purified ALK1 ECD, purity can be seen in Supplementary Fig. 8a, lane 7). The sensorgrams of BMP9, pro-BMP9, BMP10 and pro-BMP10 binding raw data (in black lines) were overlaid with a global fit to a 1:1 model with mass transport limitations (red lines). **f** A summary of kinetic parameters for ligand-receptor interactions derived from the Biacore measurements in **e**.

**a**

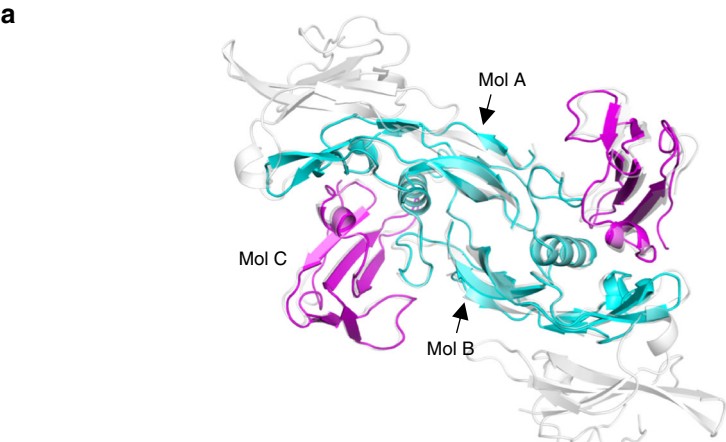

**b**

| | Buried Surface on ALK1 (Mol C) | | | Buried Surface on BMP | | |
|---|---|---|---|---|---|---|
| | Total (Å²) | % contribution from BMP Mol A | % contribution from BMP Mol B | Total (Å²) | % contribution from BMP Mol A | % contribution from BMP Mol B |
| BMP10:ALK1 | 1072.2 | 37.0 | 63.0 | 1025.5 | 32.5 | 67.5 |
| BMP9:ALK1 | 1193 | 34.6 | 65.4 | 1121.4 | 30.9 | 69.1 |

**Fig. 2 Crystal structure of human BMP10:ALK1 complex at 2.3 Å. a** Crystal structure of BMP10 (cyan) in complex with ALK1 (magenta), overlaid with the structure of BMP9:ALK1:ActRIIb complex (PDB:4FAO, in grey and semi-transparent). Mol A and Mol B are the two BMP monomers whose interfaces with ALK1 (Mol C) were analysed in **b**. **b** Comparison of the buried interface upon complex formation between BMP10:ALK1 and BMP9:ALK1 (from 4FAO). Total buried surface area and the contributing residues were calculated using PDBePISA server.

binds BMP10 in a 2:2 stoichiometry, and the two ALK1 ECD monomers are not in direct contact. The assembly of BMP10: ALK1 is identical to that of BMP9:ALK1 in the BMP9:ALK1: ActRIIb complex (PDB:4FAO), with mainchain root-mean-square deviation (RMSD) of 0.84 Å for BMP dimer, 0.52 Å for ALK1 monomer. The total buried surface area of the BMP10: ALK1 interface is slightly smaller than that of the BMP9:ALK1 interface (Fig. 2b), however, the percentage of contribution from the two monomers is similar. Importantly, the ALK1 binding sites on BMP9 and BMP10 are mostly conserved, which corresponds well with the above data that these ligands bind ALK1 with comparable affinity and stimulate ALK1-dependent signalling in endothelial cells with similar potency.

**Specificity determinants in the BMP9 and BMP10 subfamily.** The high affinity of BMP9 and BMP10 for ALK1 and their specificity for ALK1-mediated signalling suggest that these ligands possess common specificity determinants. The crystal structure of the BMP10:ALK1 complex, alongside the previously reported complex structures of BMP9[9,29,30], provide a unique opportunity to identify such specificity determinants.

Most BMPs have higher affinities for the type I receptors, such as BMP2 and BMP4 for ALK3, BMP9 and BMP10 for ALK1, and BMP14 (also called GDF5) for ALK6. The exceptions are BMP6 and BMP7, which signal via ALK2 but bind to the Activin type II receptors a and b (ActRIIa and ActRIIb) with higher affinity. Among the four type I receptors that are known to mediate BMP signalling, the high affinity ligands for ALK1, ALK3 and ALK6 have been described and crystal structures of these receptors bound to their cognate ligands been solved[9,31,32], whereas the existence of high affinity ligand(s) for ALK2 is yet to be confirmed.

A close examination of sequence alignment between the five representative BMPs with high affinities for the type I receptors as well as BMP6 and BMP7 revealed a total of 16 residues that are preferentially conserved between BMP9 and BMP10 (Fig. 3a,

highlighted in cyan, blue and yellow). Interestingly, these residues can be mapped onto three regions of BMP10 (Fig. 3b, c), the type I receptor-binding site, the type II receptor-binding site and the middle of the BMP dimer interface.

**The type I receptor binding site and conserved region 1.** Using the PDBePISA server[33], a total of 26 BMP10 residues were identified at the ALK1 binding interface (Fig. 3a, lines over the sequence). Interestingly, these residues can be divided into three groups (Fig. 4a): those conserved across different subgroups of BMPs (in red, also * over the top of the sequence in Fig. 3a), those conserved only among BMP9 and BMP10 (in cyan, referred to as conserved region 1) and those that vary between BMP9 and BMP10 (in yellow). The interactions between BMP10 and ALK1 cover the same three sites as previously identified between BMP9 and ALK1 in the 3.35 Å BMP9:ALK1:ActRIIb structure[9], but the 2.3 Å resolution reveals the BMP10:ALK1 interface in more detail (Fig. 4b–f). Sites II and III within the type I receptor binding area, which have been proposed to hold the key to ALK1 specificity for BMP9[9], indeed harbour the residues unique to BMP9 and BMP10 (Fig. 4c, d, cyan). At the site II, ALK1-specificity is defined by a hydrogen bond (H-bond) between BMP10 S339 and the mainchain oxygen of ALK1 R78, and a salt bridge between BMP10 D338 and ALK1 R80. Site III contains two residues conserved between BMP9 and BMP10 that contribute to the ALK1 specificity, but they differ in molecular interactions in the two complexes. While ALK1 H73 forms a H-bond with BMP10 P366, ALK1 E75 forms a H-bond with Y413 which is conserved across all BMPs. Overlaying the BMP2:ALK3 structure[31] revealed that although the equivalent residue Y385 in BMP2 makes a similar H-bond with ALK3 D84, there is a significant movement in BMP10 mainchain conformation due to the unique insertion of F411 (and Y416 in BMP9, Fig. 3a). Thus K368, the BMP9 and BMP10-specific residue, defines the ALK1 specificity by interacting with the mainchain oxygen on the BMP10 loop, allowing Y413 to be in close proximity for interacting with ALK1.

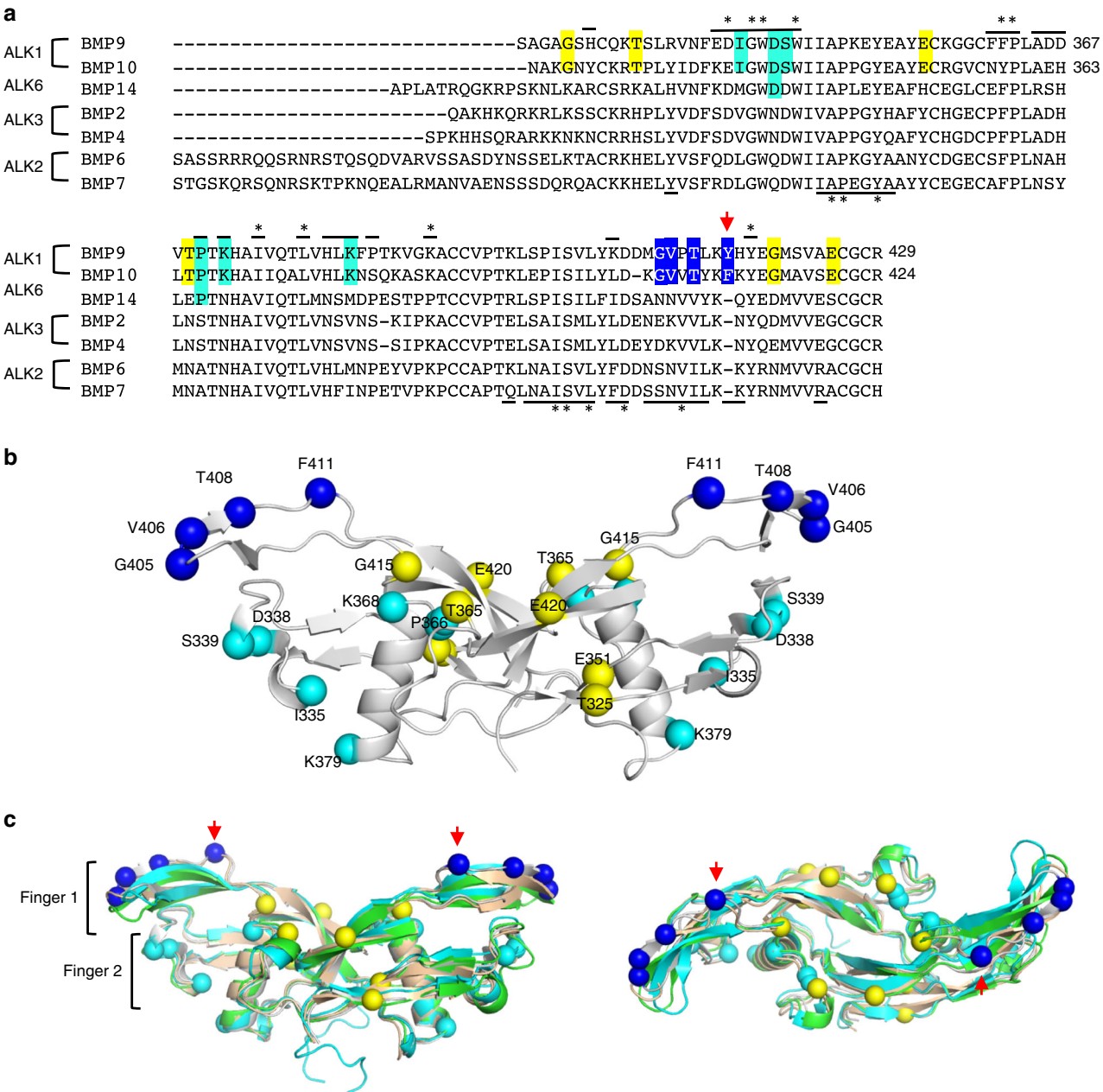

**Fig. 3 Specificity determinants in the BMP9 and BMP10 subfamily. a** Sequence alignment of representative ALK-binding BMPs. GF-domain sequences of ALK1-binding BMP9 and BMP10, ALK6-binding BMP14, ALK3-binding BMP2 and BMP4, as well as ALK2-binding BMP6 and BMP7 are aligned. Lines over and below the sequences highlight the residues at the type I and type II receptor-binding surface based on BMP10:ALK1 and BMP9:ALK1:ActRIIb structures, respectively. Asterisk (*) marks the residues that are conserved among at least 6 out of 7 aligned BMPs. Residues preferentially conserved between BMP9 and BMP10 are highlighted, in cyan for those at the type I site (conserved region 1), in blue for those at the type II site (conserved region 2) and in yellow for those outside receptor binding surface (conserved region 3). BMP10 D338 and P366 are also highlighted in cyan because they make conserved interactions with ALK1 in the crystal structure (Fig. 4). **b** Residues from conserved regions 1–3 plotted on BMP10 structure and labelled with full length proBMP10 residue numbers. Fifteen residues from conserved regions 1–3 are shown in spheres, coloured accordingly. The first Gly from conserved region 3 is not modelled in the crystal structure, and hence not plotted. **c** An overlay of BMP10 (grey) onto the structures of BMP9 (gold, from 4FAO)[9], BMP2 (green, from 2GOO)[31] and BMP7 (cyan, from 1M4U)[61] is shown from the side view (left) and the top view (right). The red arrows indicate the unique insertion in BMP9 and BMP10.

Reciprocally, ALK1 evolved a negatively charged residue with a longer side-chain at the corresponding position, E75, to allow the H-bond formation with Y413, defining its specificity for BMP9 and BMP10. The site I region contains residues that are different between BMP9 and BMP10, with the interactions predominantly mediated by sidechains (Fig. 4e). In contrast, the core of the interface is conserved across all BMPs and mostly hydrophobic in nature (Fig. 4f). From the ALK1 side, residues H73, E75, R78 and R80 interact with amino acids that are unique to BMP9 and BMP10 hence define the ALK1 specificity (Fig. 4c, d). Interestingly, these ALK1 residues are located near or within the two loops of ALK1 that have significantly different conformation

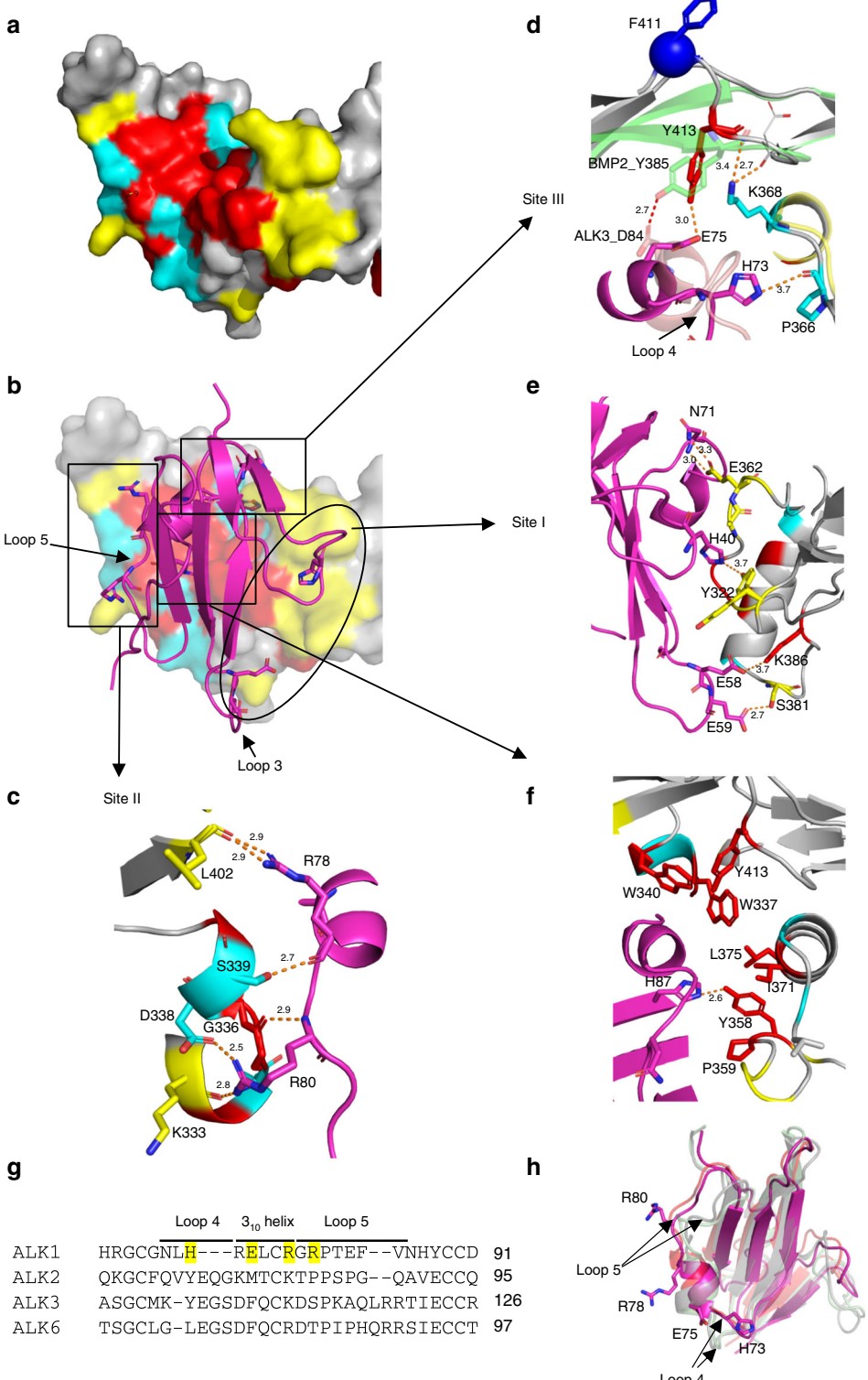

**Fig. 4 Conserved region 1 and ALK1-specificity determinants. a** ALK1-binding residues are mapped onto the BMP10 surface (grey), with those conserved across all BMPs in Fig. 3a coloured in red, those from the BMP9 and BMP10 conserved region 1 in cyan, and other variable residues in yellow. **b** ALK1 (magenta cartoon) binding to BMP10 (surface), with residues interacting with BMP10 shown in sticks. **c–f** Detailed interactions between BMP10 and ALK1. **g** Sequence alignment of four BMP-binding type I receptors, ALK1, ALK2, ALK3 and ALK6, with the four specificity-determining residues in ALK1 highlighted in yellow. Loop 4 and loop 5 are the two loops surrounding the 3₁₀ helix (Supplementary Fig. 4). **h** Overlaid structures of BMP type I receptors. The structure of ALK1 in the BMP10:ALK1 complex (magenta) is overlaid onto ALK1 in BMP9:ALK1 complex (PBD:4FAO, orange), ALK3 (PDB:2GOO, light green) and ALK6 (PDB:3EVS, light grey). ALK1 residues highlighted in **g** are shown in sticks.

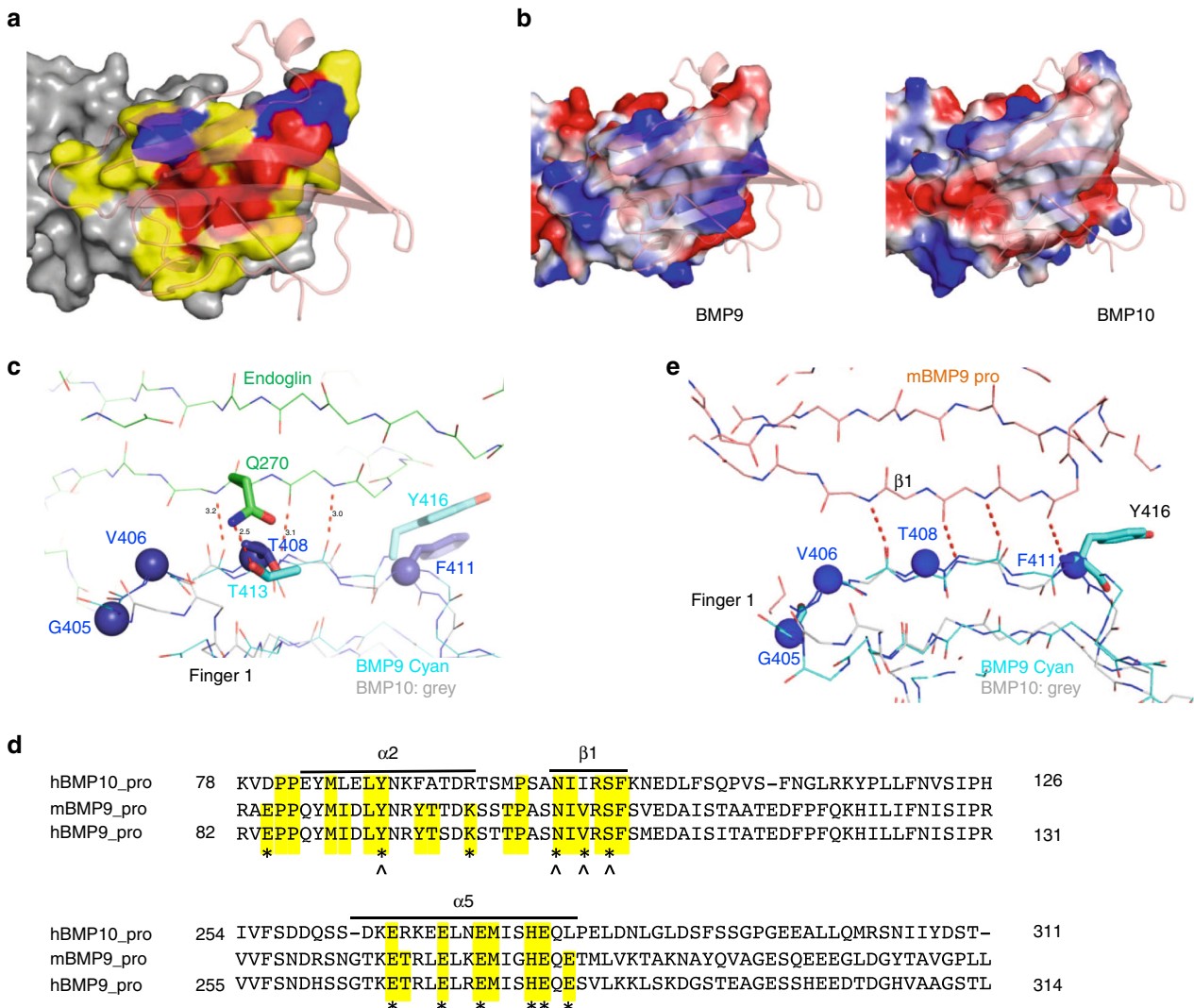

**Fig. 5 Conserved region 2 and type II site analysis. a** ActRIIb-binding residues (based on PDB:4FAO, ActRIIb in semi-transparent cartoon) are mapped onto BMP9 surface (grey), with those conserved across all BMPs in Fig. 3a coloured in red, those from conserved region 2 in blue, and other variable residues in yellow. **b** Type II binding surface of BMP9 (left) and BMP10 (right), showing as electrostatic surface (generated in PyMOL, red representing negatively charged and blue positively charged surface). ActRIIb is shown in orange, semi-transparent cartoon. **c** Residues from BMP9 conserved region 2 make three backbone β-sheet and one sidechain H-bond interactions with ENG (PDB:5HZW, ENG in green, BMP9 in cyan). BMP10 is overlaid onto BMP9 and shown in grey, with four conserved region 2 residues shown in blue spheres. Sidechains of other residues are omitted for clarity. **d** Sequence alignment of human BMP10 prodomain (hBMP10_pro) with mouse BMP9 prodomain (mBMP9_pro) and human BMP9 prodomain (hBMP9_pro). Residues at the BMP9-binding surface are highlighted in yellow and those that make direct interactions with BMP9 GF-domain are marked with *. Residues that make main chain interactions are also marked with ˆ. Only the prodomain regions that interact with BMP9 GF-domain are shown, and full-length alignment of hBMP9_pro and hBMP10_pro can be found in Supplementary Fig. 5. **e** Residues in conserved region 2 of BMP9 make four backbone H-bond β-sheet interactions with prodomain (PDB:4YCG; prodomain in orange, BMP9 in cyan. BMP10 is overlaid on BMP9 and shown in grey. Four conserved BMP10 residues are in blue spheres).

from those loops in ALK3 and ALK6 due to a 2–3 amino acid deletions (Fig. 4g, h and Supplementary Fig. 4). Therefore, ALK1 specificity is maximised through the presence of specific residues and through their unique display by shortening the length of the loops on which these residues are positioned.

**The type II receptor binding site and conserved region 2.** Of the BMP type II receptors, the structures for ActRIIa:BMP7 (PDB:1LX5)[34], ActRIIa:BMP2 (PDB:2GOO)[31], ActRIIb:BMP2 (PDB:2H62)[35] and ActRIIb:BMP9 (PDB:4FAO)[9] have been solved. Analysis of the ActRIIb binding site on BMP9 revealed that 26 residues contributed significantly to the receptor-binding interface (Fig. 3a, lines below the sequences). Similar to the type I receptor-

binding site, a subset of these residues, again mostly hydrophobic, are highly conserved across different BMPs (Fig. 3a, * below the sequences) and can be mapped to the centre of the type II surface on BMP9 (Fig. 5a, red). The variable residues, specifically those residues that differ between various BMP ligands (Fig. 5a, yellow), as well as the conserved region 2 residues that are conserved in BMP9 and BMP10 alone (Fig. 5a, blue), surround these conserved residues. When comparing the electrostatic surface, the ActRIIb binding site on BMP9 is mostly conserved in BMP10 (Fig. 5b), consistent with a previous report that both ligands bind ActRIIb with similar affinities[9].

The most striking feature in the conserved region 2 is the insertion of a hydrophobic residue, Y416 in BMP9 and F411 in

BMP10 (Fig. 3a, red arrow), located in the top strand of finger 1 (Fig. 3c, red arrows). This additional residue leads to a kink in this β-strand of BMP9 and BMP10, resulting in a novel conformation that allows BMP9 and BMP10 to form unique interactions. For example, this β-strand is in the core interaction site of BMP9 binding for ENG (Fig. 5c), making three backbone H-bonds with ENG to form an anti-parallel extended β-sheet[30]. In addition, the side chain of the conserved Thr (413 in BMP9 and 408 in BMP10) also forms a H-bond with that of ENG Q270, hence further contributing to the specificity of ENG for BMP9 and BMP10 (Fig. 5c).

Another polypeptide that binds to this conserved strand of BMP9 and BMP10 is the prodomain. PDBePISA analysis of the pro-BMP9 structure (4YCG, mouse BMP9 prodomain bound to human BMP9 GF-domain)[29] revealed that BMP9 and prodomain interactions are mostly derived from three areas of the prodomain: the α2-helix, β1-strand and α5-helix (Fig. 5d). The prodomain β1-strand makes four mainchain H-bond interactions with the conserved region 2 in BMP9 to form an extended anti-parallel β-sheet (Fig. 5e), which likely contributes to the majority of the binding energy. Since the prodomains of BMP9 and BMP10 are highly conserved, especially with respect to the contact residues in the α2-helix, β1-strand and α5-helix (Fig. 5d and Supplementary Fig. 5), the structure of pro-BMP10 is very likely to resemble that of pro-BMP9.

**Structure of pro-BMP9:ALK1 complex**. It is interesting to note that in the BMP9 prodomain, residues from the α5-helix that bind to BMP9 are exclusively charged and only make side-chain interactions (Fig. 5d). This helix interacts with BMP9 at the ALK1-binding site, and its density is missing in one of the two prodomain molecules in 4YCI[29], one of the two pro-BMP9 structures in the Protein Data Bank. This raises the question of whether ALK1 binding leads to displacement of the prodomain from BMP9. Indeed, a two-step mechanism for pro-BMP9 binding to the cell surface has been proposed[29], whereby ALK1 binding to pro-BMP9 causes the displacement of only the α5-helix and subsequent type II receptor binding leads to the complete prodomain displacement[29]. This theory would suggest the presence of a pro-BMP9:ALK1 complex in the initial stages of the signalling process. However, another report based on extensive SPR and ELISA analyses concluded that the binding of ALK1 alone displaced the entire prodomain[36]. To address this directly, we resolved the crystal structure of the human prodomain-bound BMP9 in complex with ALK1 (pro-BMP9:ALK1) to 3.3 Å, providing insight into the interaction of a prodomain-bound ligand with its high-affinity cognate receptor in the TGFβ superfamily.

The overall structure contains one copy of the BMP9 dimer, two copies of the ALK1 ECD and two copies of the prodomain. While the electron densities for BMP9 and ALK1 are excellent, the prodomains have overall poorer densities, with one copy heavily truncated and both copies missing the α5-helix (Fig. 6a and Supplementary Figs. 6, 7). Regions of the prodomain that are in direct contact with BMP9, including the α2-helix and β1-strand, have good densities (Supplementary Fig. 6b). The ALK1 and BMP9 components of this complex overlay very well with those in the BMP9:ALK1:ActRIIb complex structure (PDB:4FAO, with mainchain RMSD of 0.603 Å across BMP9:ALK1 mainchain atoms, Fig. 6b), suggesting that the BMP9 GF-domain in the pro-BMP9 complex can bind to ALK1 as a rigid body with minimal conformational change. As for the prodomain, the regions in contact with the BMP9 GF-domain (α2-helix and β1-strand) overlay well with the prodomain in 4YCG (Fig. 6c), whereas regions distal to the GF-domain are missing or overlay poorly. Overall the mainchain RMSD between the two prodomains in the

pro-BMP9:ALK1 structure is 0.533 Å, and with the two prodomains in 4YCG, 0.666–0.678 Å (note that 4YCG contains mouse prodomain which may contribute to the slightly higher RMSD). Importantly, the mainchain extended β-sheet interactions from the conserved region 2 are maintained in the pro-BMP9:ALK1 structure (Fig. 6d), suggesting that the contacts from the α5-helix may not be essential for the maintenance of BMP9:prodomain interactions.

**Solution study of the pro-BMP9 and ALK1 interaction**. To confirm that ALK1 ECD can form a ternary complex with full-length pro-BMP9, we performed an analytical gel filtration study using purified pro-BMP9 and ALK1 ECD (Fig. 6e–h). We have recently shown that soluble endoglin (sENG) can displace the prodomain effectively from the pro-BMP9 complex, with the displaced prodomain readily detectable as a different peak by analytical gel filtration[37], so we included sENG with pro-BMP9 as a control. As shown in Fig. 6e, f, pro-BMP9 and sENG alone eluted at peaks 1 and 6, respectively. When pro-BMP9 was mixed with 2-fold molar excess of ALK1, a pro-BMP9:ALK1 complex could be detected under peak 2, with excess ALK1 eluting under peak 3. The control experiment with a mixture of pro-BMP9 and sENG showed the sENG:BMP9 complex eluted under peak 4 and the dissociated prodomain eluted under a different peak (peak 5). Furthermore, Western blots of the consecutive fractions with anti-BMP9 prodomain antibody confirmed that mixing of pro-BMP9 with ALK1 did not shift the peak of the prodomain, whereas mixing pro-BMP9 with sENG led to a shift of the prodomain peak to later fractions that are smaller than 66 kDa protein marker (Fig. 6e, g). Similarly, we detected the pro-BMP10:ALK1 complex and excess ALK1 when mixing pro-BMP10 with ALK1 (Supplementary Fig. 8a). Importantly, the prodomains in the pro-BMP9:ALK1 and pro-BMP10:ALK1 complexes are in the intact form, i.e., the α5-helix is still present, as confirmed by the size of the prodomain on the SDS-PAGE (Fig. 6f and Supplementary Fig. 8a). This study confirms that ALK1 can bind to the full-length pro-BMP9 and pro-BMP10 complex without displacing the entire prodomain (Fig. 6h). A similar conclusion can also be drawn from the native gel analysis (Supplementary Fig. 8b). Pro-BMP9 runs as three bands on the native PAGE: the GF-domain (band 1), the prodomain (band 3) and the pro-BMP9 complex (band 2). Upon mixing with ALK1 ECD (band 4), newly appeared bands containing the ALK1:BMP9 complex (band 5) as well as the pro-BMP9:ALK1 complex (band 7) could be readily detected.

**CV2 does not inhibit BMP9 signalling**. The conserved region 2 corresponds to the same region where CV2 binds to BMP2 (Fig. 7a)[38]. A detailed analysis of the CV2:BMP2 structure (PDB:3BK3)[38] revealed that CV2 forms four backbone H-bond interactions with the same top strand of finger 1 in BMP2. Since BMP9 and BMP10 have an insertion and adopt a different conformation in this strand, we hypothesised that CV2 would not be able to interact with BMP9 as it does to BMP2, even though a previous publication using a reporter assay suggests that CV2 inhibits BMP9 signalling[11]. We first tested whether CV2 could inhibit BMP9 or pro-BMP9 signalling in the physiologically relevant cell type PAECs; no CV2 inhibition was observed even when CV2 was applied at a 250-fold molar excess using either a Smad1/5 phosphorylation assay (Fig. 7b) or measuring ID1 gene induction (Fig. 7c). As a positive control, we detected CV2 inhibition of BMP4 signalling in pulmonary artery smooth muscle cells (PASMCs) at 10-fold or 20-fold molar excess using the same protocol (Fig. 7d). To further confirm that CV2 is capable of inhibiting BMP2 but not BMP9 signalling, we

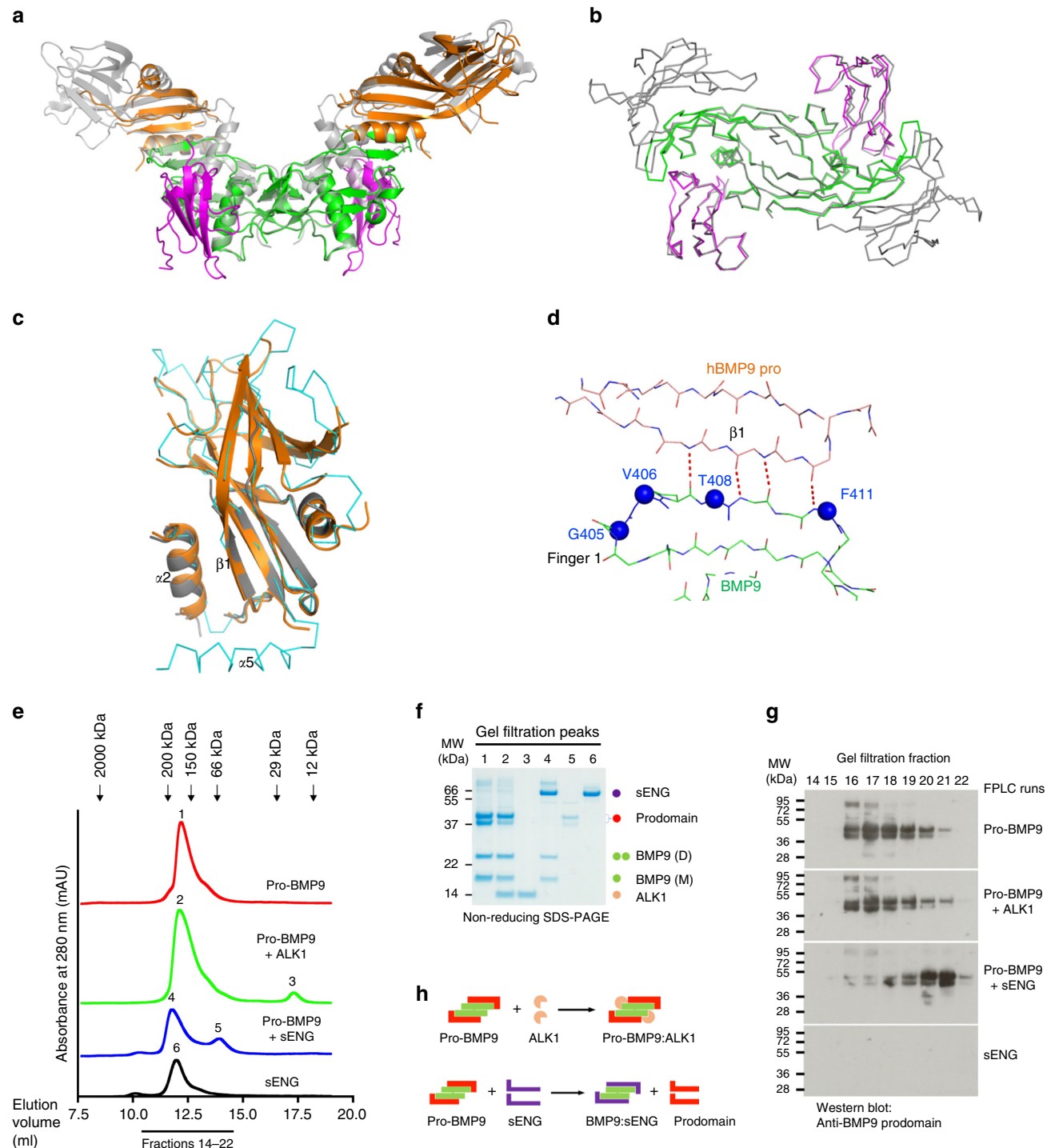

performed parallel BMP2 and BMP9 signalling assays in a single cell type, C2C12 cells, by monitoring ALP induction. As shown in Fig. 7e, both BMP2 and BMP9 induced robust ALP activity in C2C12 cells. While the presence of CV2 inhibited BMP2-induced ALP activity potently and in a dose-dependent manner, no inhibition was detected for BMP9-mediated ALP induction. These cell assays verified our structural analysis that CV2 does not inhibit BMP9 signalling.

**Modifying BMP9 signalling specificity by mutagenesis.** Our structural and sequence analyses suggest that protein–protein

recognition between BMPs and their receptors is regulated by a tripartite mechanism, whereby one set of residues are conserved across all BMPs, a second set are conserved within each BMP subfamily and the remaining residues are unique for each BMP ligand. This would suggest that the interface residues that are not conserved between BMP9 and BMP10 likely contribute to the features that vary between the two ligands. It has been shown that BMP9 is very potent in inducing ALP activity in C2C12 cells in vitro and heterotopic bone formation in vivo, whereas BMP10 does not have such activities[39]. We therefore questioned whether it is possible to generate variants of BMP9 with less or no

**Fig. 6 ALK1 can form a complex with pro-BMP9. a** Overall structure of the human pro-BMP9:ALK1 complex at 3.3 Å (BMP9 in green, ALK1 in magenta, prodomain in orange) overlaid onto the pro-BMP9 structure (4YCG, grey, semi-transparent). **b** Backbones of the BMP9:ALK1 portion from the pro-BMP9: ALK1 structure (BMP9 in green, ALK1 in magenta) overlaid onto the same region in the BMP9:ALK1:ActRIIb structure (4FAO, grey). **c** Overlay of the two prodomains from the pro-BMP9:ALK1 structure (shown in cartoon and coloured in orange and grey respectively) and that from 4YCG (in ribbon, cyan). **d** In the pro-BMP9:ALK1 structure, the conserved region 2 in BMP9 makes the same four backbone H-bond interactions with the prodomain as shown in Fig. 5e. Four residues in the conserved region 2 are shown in blue spheres and labelled with BMP10 numbering. **e–h** Analysis of complex formation by analytical gel filtration. Purified pro-BMP9, pro-BMP9 mixed with ALK1, pro-BMP9 mixed with sENG and sENG were run separately on an S200 10/300 gel filtration column which was pre-equilibrated with 20 mM Tris.HCl, 150 mM NaCl, pH 7.4. **e** Gel filtration traces. The arrows indicate the elution volumes of the standards. Numbers 1-6 indicate the 6 peaks which were analysed by SDS-PAGE. **f** Middle fraction from each peak was run on an SDS-PAGE. Identities of the proteins on the SDS-PAGE are indicated using coloured circles. **g** Consecutive fractions from each gel filtration experiment were run on a non-reducing SDS-PAGE and immunoblotted using an anti-BMP9 prodomain antibody. Each analytical sample run was repeated at least one more time with fraction checked on SDS-PAGE to ensure reproducibility. **h** Cartoon diagrams, using the same colouring scheme as the circles in **f**, to illustrate that mixing pro-BMP9 with ALK1 leads to the formation of pro-BMP9:ALK1 complex, whereas mixing pro-BMP9 with sENG leads to the displacement of the prodomain which can be readily detected as a different peak in the gel filtration.

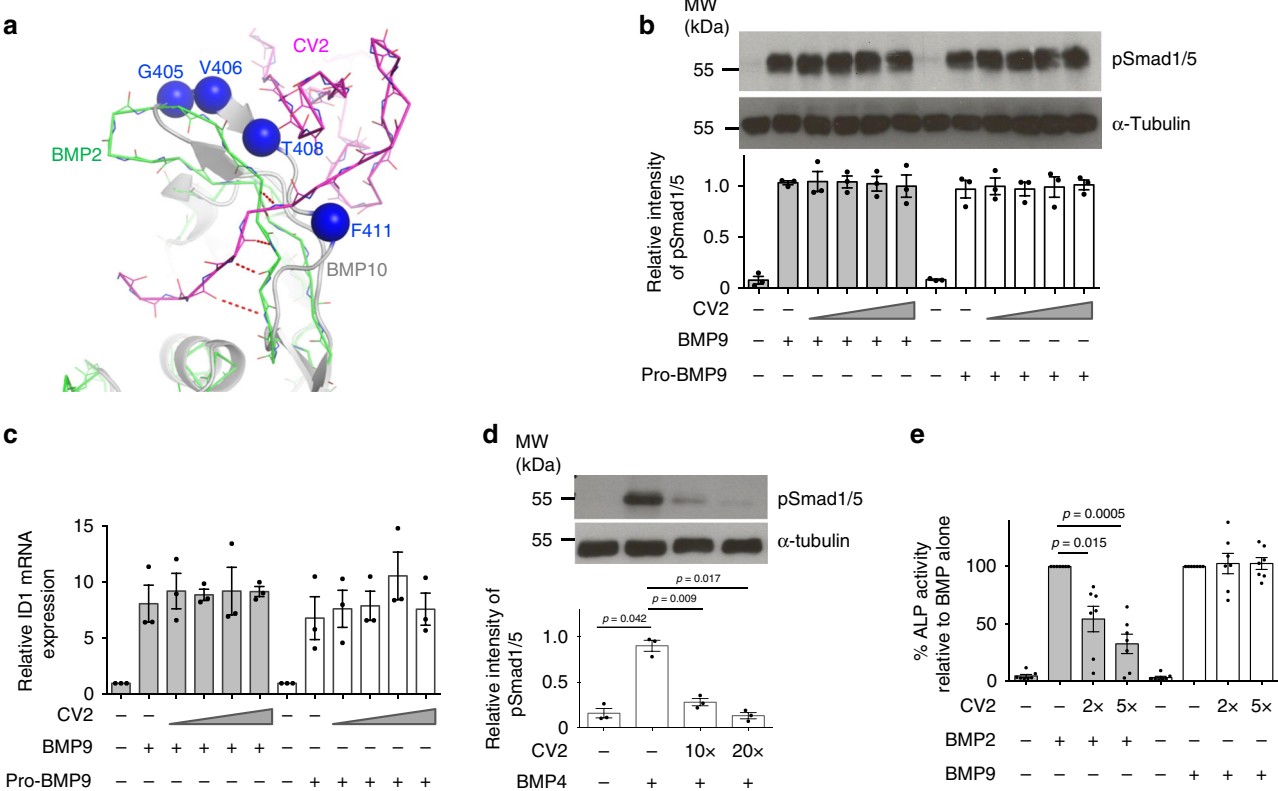

**Fig. 7 CV2 does not inhibit BMP9 signalling. a** Structural analysis. BMP10 (grey, with conserved region 2 residues in blue spheres) was overlaid onto the BMP2:CV2 structure (PDB:3BK3, CV2 in magenta and BMP2 in green). Four mainchain H-bonds that stabilise the BMP2:CV2 β-sheet interaction are shown. BMP9 has the same conformation as BMP10 in this region. **b, c** CV2 does not inhibit BMP9 signalling in PAECs. Serum-starved PAECs were treated with BMP9 or pro-BMP9 (at 1 ng ml$^{-1}$ GF-domain concentration) without or with CV2 at 10-fold, 20-fold, 50-fold or 250-fold molar excess for 15 min to assess Smad1/5 phosphorylation using immunoblots (**b**) or for 1 h to assess *ID1* gene expression using qPCR (**c**). One representative of three independent experiments is shown in **b**. Band intensity was quantified using Image J (version 1.51s). **d** CV2 inhibits BMP4 signalling in PASMCs. Serum-starved PASMCs were treated with BMP4 (25 ng ml$^{-1}$) without or with CV2 at indicated molar excess for 15 min. Immunoblots and quantification were carried out as above. $N = 3$ independent experiments and one representative blot is shown. **e** CV2 inhibits BMP2 but not BMP9 signalling in C2C12 cells. Serum-starved C2C12 cells were treated with BMP2 (130 ng ml$^{-1}$) or BMP9 (25 ng ml$^{-1}$) without or with CV2 at the indicated molar excess for 68 h. ALP activity in the cell lysate were analysed (see Methods section). $N = 7$ independent experiments. For all panels, means ± SEM are shown. **d, e** One-way ANOVA for each BMP treatment group, Dunnett's post hoc analysis against BMP alone-treated controls. Source data are provided as a Source Data file.

osteogenic activity by mutating non-conserved interface residues from BMP9 to BMP10 (those highlighted in yellow in Fig. 4a). Indeed, 4 out of the 7 mutants showed reduced ALP-induction activity in C2C12 cells compared to wild type (WT) BMP9, in particular F362Y and D366E mutations results in complete loss of BMP9-induced ALP activity under the experimental conditions

(Fig. 8a). A similar pattern was also observed when monitoring the *Id1*-gene induction by BMP9 and its variants in C2C12 cells (Supplementary Fig. 9). We further measured type I receptor binding affinities of BMP9, BMP10 and the BMP9 variant D366E which has lost ALP-induction activity in C2C12 cells. As shown in Supplementary Fig. 10, pro-BMP9, pro-BMP9 D366E and pro-

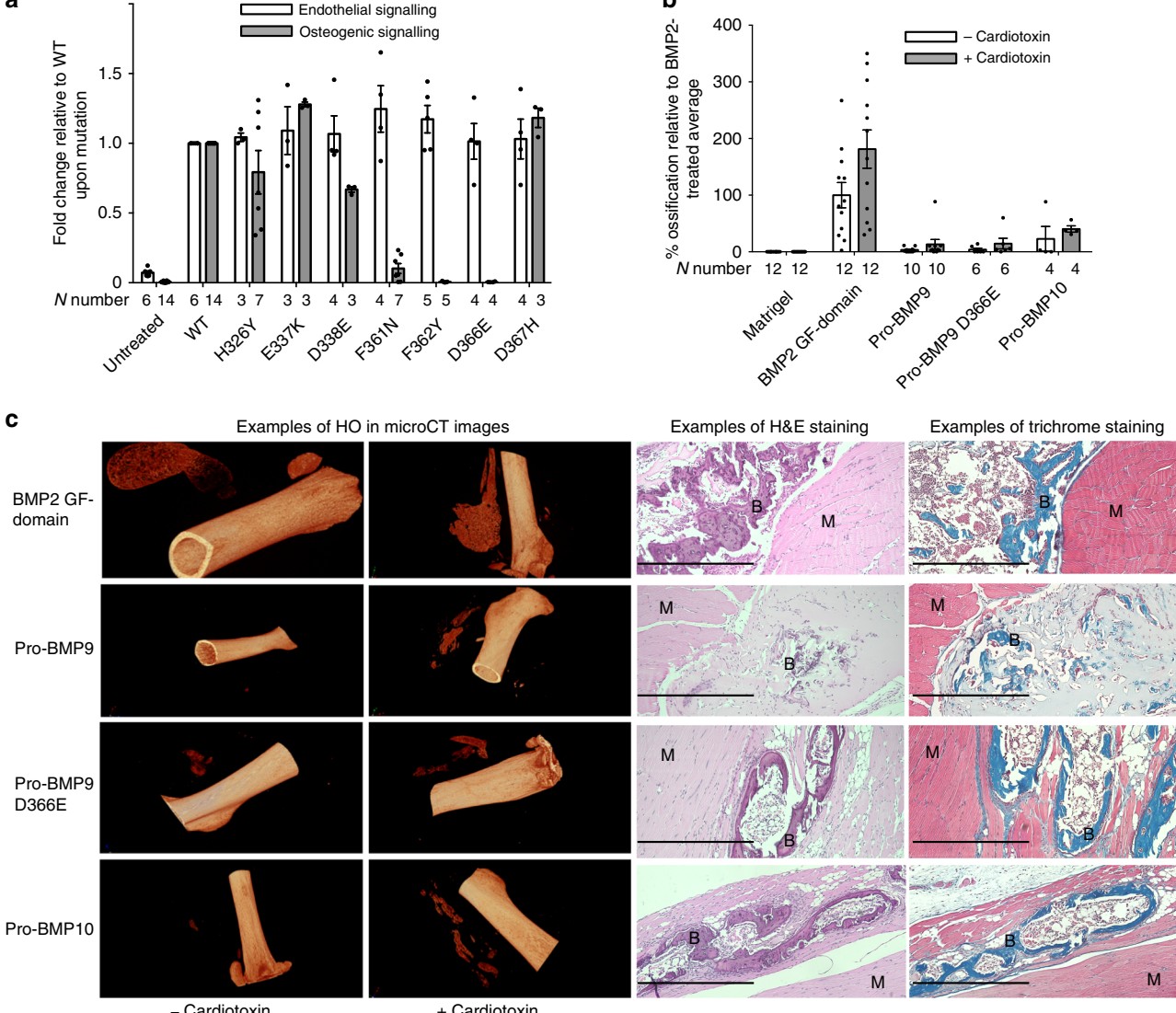

**Fig. 8 Modifying BMP9 signalling specificity by mutagenesis.** A panel of BMP9 mutants were generated, and tested in vitro and in vivo as described in the Methods. **a** Mutant proteins were subject to endothelial cell signalling assays (at 0.3 ng ml$^{-1}$ GF-domain concentration) using induction of *ID1* gene expression in hPAECs as a readout, and osteogenic signalling assays (at 10 ng ml$^{-1}$ GF-domain concentration) using ALP induction in C2C12 cells as a readout. Data were normalised to WT BMP9 and shown as fold change relative to WT upon mutation. Each treatment condition was repeated in 3–7 independent experiments alongside untreated and WT controls. The exact *N* number for each condition is given under the column. Means ± SEM are shown. **b** Recombinant WT pro-BMP9, pro-BMP9 D366E, pro-BMP10, as well as BMP2 GF-domain were subject to in vivo heterotopic bone-forming assays in the presence and absence of cardiotoxin. Each data point represents the HO result from an independent injection in one leg. *N* number for each treatment condition is given under each column. Data are presented as % ossification volume relative to the average of BMP2-treated controls. Means ± SEM are shown. **c** Representative CT images (left) and histological staining (right) of in vivo formed heterotopic bones after stimulation of indicated BMP molecules in the presence and absence of cardiotoxin. B: osteoid matrix; M: muscle cells. Scale bar = 500 μm.

BMP10 all bind to ALK1-Fc with similar affinities. But for binding to the ALK2-Fc, we can only detect the binding of pro-BMP9 and BMP9 GF-domain, not pro-BMP10 or pro-BMP9 D366E, confirming the D to E mutation at position 366 of BMP9 abolishes its ability to bind ALK2. Since BMP10 was shown to be non-osteogenic in vivo[39], we subjected pro-BMP9 D366E to an in vivo osteogenic assay, along with BMP2 GF-domain, pro-BMP9 and pro-BMP10 controls. To our surprise, in this in vivo assay with purified recombinant proteins, not only did pro-BMP9 D366E induce bone-formation, but pro-BMP10 itself was also an osteogenic BMP (Fig. 8b, c).

## Discussion

The endothelial ALK1-mediated BMP signalling pathway has attracted much attention due to its therapeutic potential for the treatment of cardiovascular disease and solid tumour. Understanding the molecular details of this pathway will facilitate the translation of therapies by improving their efficacy and specificity whilst minimising unwanted side effects. In this study, we provide essential insight into several key molecular details of the ligands involved in this signalling pathway.

Firstly, although BMP9 and BMP10 circulate in their prodomain-bound forms, most published signalling and functional assays have utilised GF-domains due to the unavailability

of pro-BMP9 and pro-BMP10 from commercial sources. We have expressed full-length human BMP9 and BMP10 in mammalian cells and purified the circulating forms of the ligands under native conditions. We demonstrate that they are functionally equivalent ligands on vascular endothelial cells in vitro and bind to ALK1 ECD with similar affinities. This provides a strong rationale that pro-BMP10 could be administered like pro-BMP9 in treating PAH[23]. However, it is worth noting that this study focused on BMP9 and BMP10 with respect to ALK1-mediated signalling in endothelial cells. At higher concentrations, both BMP9 and BMP10 can signal through non-ALK1-dependent pathways[40,41] and thus may have different signalling capacity and functional consequences.

Secondly, by solving the crystal structure of the BMP10:ALK1 complex and comparing it to published structures, we have identified three regions that are highly conserved between BMP9 and BMP10. The conserved region 1 accounts for ALK1 specificity. The conserved region 2 defines the specificity for ENG, prodomain and possibly the type II receptor as well. The conserved region 3 is located in the middle part of the BMP dimer. The crystal structure of pro-myostatin suggests that C-terminal part of the prodomain could potentially interact with myostatin GF-domain residues in this region[42]. A crystal structure of unprocessed proBMP9 or proBMP10 will be required to address this question.

The type I and type II receptor binding sites have a common feature in that each possesses residues that are conserved among all BMPs (red surface in Figs. 4a, 5a), and residues that are variable and determine BMP subgroup specificity. This is reminiscent of a well-characterised protein–protein recognition between four colicin DNases and their cognate immunity proteins. This family of closely related and structurally homologous proteins achieve specificity by a dual recognition mechanism, whereby a conserved region contributes to the major binding energy whilst a variable region modifies the interaction by further contributing positively in cognate interactions or negatively in non-cognate interactions[43,44]. There are 15 BMP molecules in humans, which signal through four type I and three type II receptors. Therefore, more complicated mechanisms are required for the regulation of specific protein–protein interactions within the BMP system. We identified a tripartite mechanism here which could explain the specificity between BMP-receptor interactions. The 16 conserved residues in the conserved regions 1–3 identified in this study distinguish BMP9 and BMP10 from other members of the BMP family, in particular those within the type I site determine the ALK1-signalling specificity.

In the crystal structure of the pro-BMP9:ALK1 complex, we observed significant proteolysis of the prodomain (Supplementary Fig. 7). Although it is possible that the excessive proteolysis was due to a crystallisation artefact, the digested regions do not make direct crystal contacts and thus there is no evidence to suggest that the proteolysis would favour crystal formation. We speculate that the binding of ALK1 to pro-BMP9 may destabilise the prodomain, rendering it more susceptible to proteolysis; hence only the regions in direct contact with BMP9 are highly ordered due to interaction restraints. Nevertheless, this structure suggests that ALK1 ECD binding to pro-BMP9 does not lead to the dissociation of the prodomain, in contrast to a previous report[36]. We further validated this conclusion using analytical gel filtration with sENG as a control[37]. Our result is in agreement with Mi et al.[29] that ALK1 binding only displace the α5-helix, not the entire prodomain from BMP9.

Structural analysis of the conserved region 2 revealed a distinct conformation adopted by BMP9 and BMP10 due to one amino acid insertion, prompted us to question whether CV2 could inhibit BMP9 and BMP10 signalling as reported previously[11,12]. Using a range of cell-based signalling assays, we have conclusively

shown that CV2 does not inhibit BMP9 signalling. This observation has important translational implications; for instance, if BMP9 is to be developed into a therapy for PAH, the presence of any circulating ligand traps must be taken into account.

Finally, during the validation of the structural findings, we performed mutagenesis of BMP9 to modify its specificity and evaluated its osteogenic activity in vitro and in vivo. We found unexpectedly that BMP10 is also an osteogenic BMP that can promote heterotopic bone-formation in vivo, in contrast to a previous report[39]. This is likely due to limitations in the methodology employed in the previous study, whereby C2C12 cells infected with adenoviral constructs containing the BMP10 gene were used for the ALP activity and in vivo bone-forming assay. It is not known in that study whether the BMP10 produced from transfected C2C12 cells was fully processed and active. In our study, we used purified, fully processed recombinant pro-BMP10 with activity validated in PAECs[5]. Importantly, our data also demonstrate that lack of ALP activity in vitro does not necessarily predict lack of osteogenic activity in vivo. More studies are required to understand molecular mechanisms that determine the bone-inducing activity of BMP9 and BMP10.

In summary, using a combination of signalling assays, protein biochemistry and X-ray crystallography, we provide extensive and detailed molecular insights into ALK1-mediated signalling by circulating forms of BMP9 and BMP10. Such knowledge will provide essential information for improving the specificity and efficacy of therapies that target the endothelial BMP signalling pathways.

## Methods

**Materials**. Rabbit antibody against phospho-Smad1/5 (pSmad1/5, cat. No. 9516) was purchased from Cell Signalling Technology. Mouse monoclonal anti-α-tubulin (cat. No. T6199) was purchased from Sigma-Aldrich. Secondary anti-rabbit horseradish peroxidase (HRP) antibody (cat. No. P0448) and anti-mouse HRP antibody (cat. No. P0447) were both from Dako. Plasmid purification kits and RNA extraction kits were purchased from Qiagen. DMEM and CDCHO media were purchased from Thermo Fisher Scientific. Pulmonary artery endothelial cells (PAECs, cat. No. CC-2530) and endothelial growth medium (EGM2) were purchased from Lonza. HEK EBNA cells (cat. No. CRL-10852) and C2C12 cells (cat. No. CRL-1772) were purchased from ATCC. Chromatography columns were purchased from GE Healthcare. All crystallisation reagents were purchased from Hampton Research Inc and Molecular Dimensions. Human GF-domains of BMP2, BMP9 and BMP10, CV2, ALK1-Fc and ALK2-Fc were all purchased from R&D Systems.

**Expression and purification of the ALK1 ECD**. Human ALK1 cDNA (NM_000020) encoding amino acids 22-118 was cloned into pET39b (70090, Novagen) between NcoI and NotI sites to create a fusion protein DsbA-(His)₆-ALK1 ECD. A TEV (Tobacco Etch Virus nuclear inclusion A endopeptidase) protease cleavage site was introduced at the N-terminus of the ALK1 ECD to facilitate release of the untagged ECD. The construct was confirmed by DNA sequencing and transformed into bacterial strain Rosetta DE3 (70954, Novagen) for protein expression. Cells were grown to mid-log phase followed by isopropyl β-D-thiogalactopyranoside induction and further incubation at 22 °C overnight. ALK1 ECD was purified from periplasmic fractions following the method described previously for the BMPRII ECD[5]. Briefly, total periplasmic proteins were extracted following the pET System Manual (Novagen) and His-tagged fusion proteins were purified on a 5 ml nickel-nitrilotriacetic acid column (GE Healthcare). Fractions containing the fusion protein were pooled, dialysed into Tris Buffered Saline, and incubated overnight with His-tagged TEV protease. The mixture was passed through a pre-charged nickel-nitrilotriacetic acid column to remove His-tagged DsbA and TEV protease. ALK1 ECD, which was in the flow-through and wash solution, was concentrated and further purified by gel filtration on a Superdex 75 column.

**Expression and purification of pro-BMP9 and pro-BMP10**. Full-length human BMP9 cDNA was cloned into pCEP4 (V04450, Invitrogen) between HindIII and XhoI sites and verified by DNA sequencing. Cloning of full length human pro-BMP10, transfection and purification of pro-BMP9 and pro-BMP10 were achieved following previously described protocols[5]. Briefly, pCEP4 constructs containing either proBMP9 or proBMP10 were purified using Qiagen midi prep kit. HEK EBNA cells were seeded in ten triple flasks and grown to 50–80% confluency before being transfected using polyethylenimine. Serum-free CDCHO medium was

applied from Day 2 and conditioned media were harvested every 3–4 days over 2 weeks. To facilitate the processing, human full-length furin cDNA, also cloned in pCEP4, was co-transfected with proBMP10.

To purify pro-BMP9 or pro-BMP10, 5 l of conditioned medium was loaded onto two tandemly linked 5 ml Hitrap Q columns, pre-equilibrated in 20 mM Tris. HCl, pH7.4. After elution with NaCl gradient, fractions containing pro-BMP9 or pro-BMP10 were identified on a non-reducing SDS-PAGE, pooled, concentrated using VivaSpin columns (Sartorius) and loaded onto a HiLoad 16/600 Superdex 200 pg gel filtration column pre-equilibrated in 20 mM Tris.HCl, 50 mM NaCl. Fractions containing target proteins were identified by SDS-PAGE and Coomassie staining, further purified on a MonoP 5/200 GL column which was pre-equilibrated in 20 mM Tris.Cl, pH7.4. After elution with NaCl gradient, target proteins were pooled and subject to a final purification step on a Superdex 200 column, pre-equilibrated in 20 mM Tris.HCl, pH7.4, 150 mM NaCl.

**Expression and purification of sENG.** The expression and purification of sENG have been described recently[37]. Briefly, residue 1–586 of human ENG cDNA was cloned into pCEP4 with a C-terminal 6xHis tag and sequence was confirmed by DNA sequencing. Plasmids containing sENG were transiently transfected into HEK EBNA cells in DMEM containing 5% FBS before changing into CDCHO expression media. Conditioned medium was harvested every 3–4 days for up to 2 weeks and used for purification. To purify sENG, 5 l of conditioned medium were loaded onto a 5 ml HiTrap Excel column (GE Healthcare) pre-equilibrated in Buffer A (50 mM TrisHCl, pH7.4, 500 mM NaCl, 5 mM imidazole). After wash, sENG was eluted with 5–1000 mM imidazole gradient in buffer A. Fractions containing sENG were identified on SDS-PAGE, pooled and dialysed against 50 mM Tris.HCl, pH7.4, 50 mM NaCl and then loaded onto a 5 ml HiTrap Q HP column. After eluting with a NaCl gradient, fractions containing sENG were identified by non-reducing SDS-PAGE, pooled and further purified on a Superdex 200 16/600 gel filtration column.

**Endothelial cell signalling assays.** Human PAECs were grown in EGM2 with 10% FBS until 90% confluent before undergoing quiescence for 16 h in EGM2 with 0.1% FBS. Cells were treated with GF-domains of BMP9 and BMP10, or pro-BMP9 and pro-BMP10 with detailed conditions described in legends to Figs. 1, 7, 8. Immunoblotting and qPCR analyses were carried out as detailed below.

**Immunoblotting for detecting Smad1/5 phosphorylation.** Cell lysates were quantified using Detergent Compatible (DC)™ Protein Assay Kit II (cat. No. 5000112; Bio-Rad) following which 40 µg of protein was loaded and fractionated on a 10% SDS-PAGE. After transfer and blocking, the membranes were incubated with anti-pSmad1/5 antibody (1:300 dilution) followed by wash and secondary anti-rabbit HRP antibody (1:2000 dilution). After further wash and detection with chemiluminescence reagents (GE Bioscieince), the blots were washed in PBS and re-probed with mouse anti-α-tubulin antibody (1:2000 dilution). After washing and incubating with secondary anti-mouse HRP antibody (1:2000 dilution), the target protein was revealed using chemiluminescence kit.

**RNA extraction and qPCR analysis.** At the end of the treatment, cells were washed and snap-frozen. Total RNA was extracted using RNeasy Plus Mini kit and reverse transcription was carried out using High-Capacity cDNA Reverse Transcription Kit (Applied Biosystems) following manufacturer's instruction. The following primers were used for the qPCR analysis: human ID1: 5′-CTGCTCTACGACATGAACGGC, 5′-TGACGTGCTGGAGAATCTCCA; human β-2-microglobulin (B2M): 5′-CTCGC GCTACTCTCTCTTTCT, 5′-CATTCTCTGCTGGATGACGTG; The reactions were carried out on a StepOnePlus™ cycler (Applied BioSystems) and the relative expression of ID1 was normalised to B2M using the ΔΔCT method, shown as fold-change relative to control samples.

**Microarray analysis.** Four human PAEC lines were grown as described above and treated with pro-BMP9 or pro-BMP10 (both at 0.3 ng ml⁻¹ GF-domain final concentration) for 1.5 h before cells were harvested for mRNA extraction. Microarray experiments were performed at Cambridge Genomic Services, using a human Gene 2.1 ST Array Plate (Affymetrix, Wooburn Green, UK) in combination with WT PLUS amplification kit (Affymetrix) according to the manufacturer's instructions. Following data processing using package Oligo (version 1.40.2) in R (version 3.3)[45], normalisation was carried out using Robust Multichip Analysis (RMA)[46], and comparisons were performed using the limma package (version 3.32.1)[47]. The results were corrected for multiple testing using false discovery rate (FDR).

**Surface plasmon resonance analysis.** Receptor-ligand binding experiments were performed using the Biacore T100 biosensor (Biacore/GE Healthcare). ALK1-Fc or ALK1 monomer was immobilised onto a Series S research grade CM5 sensor chip by amine-coupling at a density of 800 or 270 resonance units, respectively. Flow cell 1 was used as the blank reference cell for subtraction during analysis. For kinetic measurements, a concentrated series of BMP9 GF-domain, pro-BMP9, BMP10 GF-domain or pro-BMP10 were injected in duplicate over the flow cells at

a flow rate of 30 µl min⁻¹ in a buffer containing 0.01 M HEPES, pH7.4, 0.5 M NaCl, 3 mM EDTA, 0.5 mg ml⁻¹ BSA and 0.01% (v/v) Surfactant P20 at 25 °C. The binding surface was regenerated with 4 M Guanidine Hydrochloride between each cycle by four short injections (8 s each) at a flow rate of 30 µl min⁻¹. The kinetic rate constants were obtained by fitting the corrected data to a 1:1 interaction model using Biacore T100 Evaluation Software (version 2.0.3, GE Healthcare). The equilibrium binding constant $K_D$ was determined by the ratio of binding rate constants $k_d/k_a$.

**Structure of the BMP10:ALK1 complex.** To purify the BMP10 GF-domain, pro-BMP10 was denatured in 7 M urea solution overnight before being loaded onto a 5 ml HiTrap S column equilibrated in a buffer containing 25 mM NaCl, 6 M urea and 20 mM Tris pH 7.4. Bound proteins were eluted with a 25–1000 mM NaCl gradient in 6 M urea, 20 mM Tris pH 7.4. Fractions containing BMP10 GF-domain disulphide-linked dimer were concentrated to 1 ml, then rapidly diluted with 19 ml of cold refolding buffer (1 M NaCl, 10% glycerol, 3% CHAPS, 2.5% glycine, 5 mM glutamic acid, pH4.0), and left to refold on a roller at 4 °C for 10 days. The refolded mixture was subject to centrifugation and the supernatant containing soluble, refolded BMP10 was recovered. Excess of purified ALK1 ECD was added directly to BMP10 in the refolding buffer, further incubated at 4 °C overnight. The mixture was then concentrated to 1 ml and applied to a Superdex 200 16/600 column pre-equilibrated in 20 mM Tris pH 7.4 and 200 mM NaCl. Fractions containing BMP10:ALK1 ECD were identified by SDS-PAGE and Coomassie staining, pooled and concentrated to 1.2 mg ml⁻¹ for crystallisation. Hanging-drop crystallisation trials, using 1 µl protein and 1 µl reservoir solution, were set up and diffraction quality crystals were obtained over 1 month in 17% PEG 3350, 0.175 M NH₄I, 0.1 M HEPES pH 7.8 at 22 °C. Crystals were cryo-protected in 30% glycerol, 19% PEG3350, 0.175 M NH₄I, 0.1 M HEPES pH 7.8 and flash-frozen in liquid nitrogen. Data were collected at 100 K at Diamond Light Source (DLS) on Beamline I03 at a wavelength of 0.9763 Å from a single crystal, and processed in space group $P6_522$ to 2.8 Å using iMOSFLM[48], Aimless[49] and Scala[50] in CCP4 suite (version 7.0-macosx-x86_64)[51]. The structure was solved by molecular replacement using Phaser[52], with BMP9 and ALK1 in 4FAO in the search models. Model building was carried out using Coot[53], and refined using REFMAC5[54] and phenix.refine (version 1.11.1_2575)[55]. Validation was performed using MolProbity[56]. A second crystal form was subsequently obtained from Morpheus 2 screen condition 48 diluted with 30% water, which includes 0.07 M Amino acids (0.014 M DL-Glutamic acids monohydrate; 0.014 M DL-Alanine; 0.014 M Glycine; 0.014 M DL-Lysine mono-hydrochloride; 0.014 M DL-Serine), 0.07 M Buffer System 3 (0.07 M Tris(base)/ BICINE) pH 8.5, 35% Precipitant Mix 4 (9% v/v MPD, 9% w/w PEG1000, 9% w/ PEG3350). Data collection was carried out at I03 (DLS) from a single crystal. Data was processed by DIALS[57] and aimless in space group $P6_522$ to 2.3 Å. The structure was solved by molecular replacement using Phaser, with the 2.8 Å BMP10:ALK1 ECD structure as the search model. Model building, refinement and validation were carried out as above. The dihedral angles of 96.59 and 97.16% of all amino-acid residues for the 2.3 and 2.8 Å structures are in the favoured region and none in the disallowed region. All the data collection, data reduction, structure determination and refinement statistics are shown in Supplementary Table 1. The coordinates were deposited in the PDB with the access codes 6SF1 (2.8 Å) and 6SF3 (2.3 Å). The two structures are almost identical, with small changes in some loop area (Supplementary Fig. 2). The 2.3 Å resolution structure was used in the structural analysis in the Results section.

**Structure of the pro-BMP9:ALK1 complex.** Purified pro-BMP9 was mixed with ALK1 ECD in a 1:1.5 molar ratio and incubated for 30 min at room temperature, following which the mixture was loaded onto a Superdex 200 16/600 column pre-equilibrated in 150 mM NaCl, 20 mM Tris pH 7.4. Fractions containing the pro-BMP9:ALK1 ECD complex were identified by SDS-PAGE and Coomassie staining, combined and concentrated to 10 mg ml⁻¹ to set up crystallisation trials in hanging-drops, using 1 µl protein and 1 µl reservoir solution. Diffraction quality crystals were obtained by micro-seedings in 0.14 M K/Na tartrate, 14% PEG 3350 over 4 weeks at 22 °C. Data were collected at 100 K from a single crystal, cryo-protected in 25% glycerol in 0.14 M K/Na tartrate, 16% PEG 3350, at DLS on I04-1 at a wavelength of 0.9282 Å, processed in $P6_1$ to 3.3 Å. The structure was solved by molecular replacement using the BMP9:ALK1 structure from 4FAO and mouse prodomain from 4YCG as search models. Data processing, molecular replacement, model building and refinement were carried out using the software packages as described above. The dihedral angles of 95.72% of all amino-acid residues are in the favoured region and none in the disallowed region. The coordinates were deposited in the PDB with the access code 6SF2. All the data collection, data reduction, structure determination and refinement statistics are shown in Supplementary Table 1.

**Structural analysis and sequence alignment.** Structural analyses were performed and figures generated using PyMOL (The PyMOL Molecular Graphics System, Version 1.2r3pre, Schrödinger, LLC). For BMP10:ALK1 structure, the 2.3 Å structure was used for generating the final figures. Sequence alignments were performed using Clustal Omega[58]. Protein–protein interaction interfaces were analysed using the PDBePISA server (https://www.ebi.ac.uk/pdbe/pisa/)[33]. The

numberings for all proteins in the alignments and structures begin at the start of the open reading frame i.e., Met1 of the signal peptide.

**Analytical gel filtration**. Purified pro-BMP9, sENG or pro-BMP10 (all at 100 μg) was mixed with a 2-fold molar excess of purified ALK1 ECD (in 150 mM NaCl, 20 mM Tris pH 7.4) or buffer control to a final volume of 200 μl. Above mixtures, alongside ALK1 ECD control, were run on a Superdex 200 10/300 size-exclusion column and 250 μl fractions were collected. Peak fractions from each run were assessed by SDS-PAGE, Coomassie or silver-staining. Traces were exported from AKTA Unicorn 6 control software, then aligned and displayed using Prism 6 software.

**PASMC isolation**. PASMCs were obtained from the lung resection specimens from the Papworth Hospital Research Tissue Bank. Ethical approval was obtained from the NRES Committee East of England -Cambridge East (reference number: 18/EE/0269). Informed consents were obtained from all donors. Isolation of PASMCs has been described previously[59]. Briefly, lung sections containing small vessels (<2 mm diameter) were obtained. The lung parenchyma is dissected away and small vessels were cut into small pieces around 0.2 cm × 0.2 cm. The chopped vessel pieces were transferred to a T25 flask in DMEM medium supplemented with 20% foetal calf serum. Allowing approximately two weeks for cells to attach and expand before passage. PASMC cell type was confirmed by the characteristic elongated morphology and immunostaining with calponin antibody. A negative mycoplasma test was obtained before the cells were used for the signalling assays. The PASMCs isolated from three individuals were used in the signalling assays.

**Signalling assays in PASMCs**. PASMCs were cultured in DMEM media with 10% FBS and antibiotics until 90% confluent before undergoing quiescence in DMEM with 0.1% FBS for 48 h. Cells were treated with BMP4 GF-domain (25 ng ml$^{-1}$) in the presence or absence of CV2 at indicated concentrations for 15 min followed by protein extraction and pSmad1/5 immunoblotting as described above.

**C2C12 cell signalling assays**. C2C12 mouse myoblast cells were cultured in DMEM media supplemented with 10% FBS and antibiotics until about 70% confluent. Treatment mixture containing BMP2 or BMP9 with or without CV2 at indicated concentrations were made up to 1 ml with DMEM containing 0.25% FBS and antibiotics, followed by 30 minutes incubation at room temperature. The cells were washed twice with 1 ml DMEM containing 0.25% FBS before treatment mixture applied. After 68 h, cells were snap-frozen using a dry ice ethanol bath and lysed with cold lysis buffer (1% Triton X-100 in PBS) and two freeze-thaw cycles. The cell lysates were collected and centrifuged at 17,000 × g for 5 min at 4 °C to remove insoluble material. Protein quantification was performed on the cell lysates using the DC™ Protein Assay Kit II (BioRad). ALP activity was measured using a 96-well plate. Duplicate wells were prepared with 20 μg of cell lysate protein in a total of 50 μl lysis buffer before 100 μl of pNPP substrate was added (Sigma-Aldrich) in 0.1 M glycine buffer, pH 10.4, containing 1 mM MgCl$_2$ and 1 mM ZnCl$_2$. The plate was covered in foil and incubated at 37 °C for 30 min, following which the absorbance was periodically read at 405 nm on a Microplate Reader.

**BMP9 mutagenesis and in vitro signalling assays**. A panel of BMP9 mutants was generated by replacing specified residues at the type I receptor site with equivalent residues from BMP10 using site-directed mutagenesis. Mutations were verified by DNA sequencing. Mutant proteins, alongside WT pro-BMP9, were expressed in HEK-EBNA cells in T25 flasks. Because mutations in the BMP9 GF-domain may change the antibody binding epitope, current published ELISA methods which use antibodies against the GF-domain are not suitable for quantifying the mutant proteins in the transfected medium. On the other hand, pro-domain and the GF-domain are produced from the same polypeptide chain and processed in a 1:1 ratio, hence the ratio of prodomain of BMP9 mutant to WT in the conditioned media will be identical to the ratio of mature GF-domains. We therefore quantified Pro-BMP9 WT and mutant proteins in the conditioned media from each transfection following a previously published method as below[28]. The conditioned media were subject to two repeats of anti-prodomain Western blots and the band intensity of mutants to WT calculated. The concentration of WT BMP9 in the transfection medium was determined by ELISA using BMP9 from R&D Systems as standards. The concentrations of the mutant proteins were calculated using the ratio obtained from anti-prodomain blots. Endothelial cell and C2C12 cell signalling assays were carried out as detailed above. Fold changes relative to the WT were quantified from three independent HEK-EBNA transfection and quantification experiments. Means ± SEM are shown.

**In vivo heterotopic ossification (HO) assay**. All in vivo animal studies were performed in accordance with the UK Home Office regulations (Project Licence: PPL PAE2D0A13) and ethical approval was obtained from the University of Cambridge's Animal Welfare and Ethical Review Board.

Male C57/BL6 mice (Charles Rivers UK Ltd) of 7–8 weeks' old were housed in groups of four in individually vented cages with free access to food and water. After an acclimatisation period of 10 days, they were anaesthetised with isoflurane in O$_2$ before the implantation of reduced growth factor Matrigels (Corning, Growth Factor Reduced, cat. No. 356231) loaded with purified pro-BMP9, pro-BMP9 D366E, pro-BMP10 or BMP2 GF-domain (from R&D Systems) into the mid belly of the Quadriceps Femoris (QF) muscle. Because BMP9-induced heterotopic bone formation requires local muscle damage which could be achieved by cardiotoxin injection, we performed the bone formation assay in the presence and absence of cardiotoxin[60]. Cardiotoxin (40 μl 10 μM from Naja Pallida, Latoxan, Portes les Valence, France. cat. No. L8102) was injected two days prior to the Matrigel/BMP implant in to the QF of one leg to induce local inflammation. Ex vivo morphological analysis of heterotopic bone formation was carried out 14 days after the BMP application by microCT (Bruker, SkyScan 1172). 3D MicroCT images (Fig. 8c) were reconstructed from 180 radiograph projections using filtered back projection in CTvox software (Bruker). The HO volume was calculated from contiguous 5 μm thick coronal CT slices. The HO area per slice was measured by importing the CT sections into ImageJ and the HO segmented from the surrounding muscle and bone by lower and upper threshold techniques. The HO volume was calculated from the measured area and the interslice thickness. Sections of the tissues containing the heterotopic bone were examined by Hematoxylin & eosin staining and Masson's trichrome staining.

**Reporting summary**. Further information on research design is available in the Nature Research Reporting Summary linked to this article.

## Data availability
Coordinates and structure factors for all structures have been deposited to the Protein Data Bank, with the accession numbers of 6SF1, 6SF2 and 6SF3. Microarray data have been deposited to Gene Expression Omnibus, with the accession number of GSE134890. The source data underlying Figs. 1a, 7b–e, 8a, b and Supplementary Fig. 9 are provided as a Source Data file. All additional experimental data are available from the corresponding author on request.

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

## Acknowledgements

This work was supported by the British Heart Foundation grants PG/12/54/29734, PG/15/39/31519 and PG/17/1/32532 to W.L. and N.W.M. We would like to thank Prof. Peter ten Dijke for constructive comments, Diamond Light Source for beamtime (proposals MX11235 and MX15916), and the support by the staff on beamlines I03 and I04-1 for assistance with crystal testing and data collection. We also thank the support from Cambridge Genomic Service, in particular Dr. Emily Clemente and Dr. Julien Bauer, during the microarray data analysis.

## Author contributions

R.M.S., J.G., J.H.W., Z.T., J.S.B., A.L., and M.Y. collected and analysed the data. R.M.S., D.J.G., J.R., N.W.M., and W.L. designed the experiments and analysed the data. W.L. performed structural and sequence analyses, coordinated the study and drafted the manuscript. All authors contributed to the writing of the paper.

## Competing interests

N.W.M. and W.L. are co-founders of Morphogen-IX. D.J.G. and J.R. are co-founders of RxCelerate Ltd. J.S.B. and J.R. are full-time employees of RxCelerate Ltd. The remaining authors declare no competing interests.
