## [Peer Review File · Nature Communications]

Reviewers' comments:

Reviewer #1 (Remarks to the Author):

Salmon RM et al investigated the molecular and structural bases of ALK1 in mediating BMP9/BMP10 signaling activities. By analyzing the crystal structures, the authors showed that a tripartite recognition mechanism of BMP10:ALK1 complex defined BMP9 and BMP10 specificity, and that the prodomain-bound BMP9:ALK1 complex represented the circulating BMP9's initial encounter with the high-affinity receptor. Furthermore, structural and experimental evidence did not support Crossveinless 2 as a BMP9 inhibitor. Interestingly, the introduction of BMP10-specific residues into BMP9 abolished BMP9-induced ALP in C2C12 cells, but did not affect BMP9-induced osteogenic activity in vivo. The authors further demonstrated that BMP10 also exhibited osteogenic activity in vivo, which is rather unexpected.

The reported findings are interesting and shed important insights into the role of ALK1 in mediating BMP9 vs. BMP10 signaling. The findings also provide structural evidence about the distinct signaling mechanisms regarding the signal. The reported work represents a significant undertaking. The experimental design was straightforward and reasonably executed. For the most part, data analysis and interpretations were appropriate. While the structural analysis of the BMP-ALK1 interactions was fairly thorough and well carried out, some biological aspects of the studies were not conceived, which might have led to certain unexpected findings. Thus, the manuscript will be significantly improved if the following concerns are fully addressed:

- 1). The commercially available recombinant BMP9 and BMP10 from R & D Systems were used in this study. While these proteins have been used for numerous published studies and may be suitable for protein structure studies, the biological activities of these recombinant proteins may be comprised and are only tested in a limited fashion, i.e., ALP activity in MC-3T3 osteoblastic cells and/or ATDC5 chondrogenic cells. Thus, the validity of the study, especially regarding reported biological (osteogenic) activities of BMP9 vs. BMP10, is a concern. According to the vendor's description, some of the rhBMPs are tagged with 6xHis at C-termini, which may be detrimental to their biological activities. In fact, this may explain some of the inconsistent observations reported by the authors: rhBMP9 was less effective in inducing osteogenic signaling than that of BMP2; BMP9 with defective ALP activity was still osteogenic in vivo; and rhBMP10 was shown to exhibit osteogenic activity in vivo, which has never been observed in other studies. In order to more definitively resolve these unexpected findings, the authors should exogenously express the BMP9 mutants and BMP10 in osteoblast progenitors and verify their osteogenic activities in vitro and/or in vivo.

- 2). The biological activities of BMP9 and BMP10 (and their derivatives) were tested in one line, C2C12, which is a myoblastic line and not a prototypic osteoblast progenitor line. Other mesenchymal stem cell (MSC) lines or primary MSCs should be used to validate the findings.

- 3). As BMP9 and BMP10 can form heterodimers, it's conceivable that the observed BMP10-induced osteogenic activity may be caused by BMP10 "hijacking" BMP9 activity. Thus, BMP9-null or silenced MSCs may be used to determine whether BMP10 is osteogenic per se and/or BMP9-independent.

- 4). Along the same line, it's conceivable that BMP10 may exhibit osteogenic activity in osteoblastic differentiation stage-dependent fashion; e.g., higher (or detectable) osteogenicity in more differentiated bone progenitors, but less active in early stage of MSCs. Thus, the authors should again test BMP10 in different bone progenitor cells.

- 5). The authors found CV2 did not inhibit BMP9 signaling in C2C12 cells. Based on the structural analysis, can the authors predict how effectively BMP antagonists such as noggin and gremlin would inhibit BMP9 or BMP10?

- 6). In Figure 8, it was not clear why Cardiotoxin was used to stimulate in vivo HO. The retrieved HO bony masses should be examined by H & E and trichrome staining to verify heterotopic ossification.

- 7). Minor comments: PAECs and PSMCs were not defined in the text in their first appearance.

Reviewer #2 (Remarks to the Author):

The authors illuminated the molecular mechanisms of BMP9/BMP10-ALK1 signaling using a wide range of methods including crystallography, biophysical binding assays, cellular and in vivo mouse studies. They reveal a rather unexpected role of the pro-domains of BMP9/10 and their passive role in signalling and binding to ALK1. The authors have performed detailed structural analyses of BMP10-ALK1 and pro-BMP9-ALK1 complexes. These structural analyses then shed light on our understanding of the role of CV2 and guided site-directed mutagenesis of BMP9/10, cellular and in vivo assays to decipher BMP9/10 specificity. The results are of high quality, timely and of high interest to a wide community of structural and developmental biologists, cancer researchers and drug discoverers due to a wide range of mechanisms controlled by BMP9/10 in human health and disease.

However, I have one major concern, which would need addressing - the co-existence of the pro-BMP9-ALK1 complex. The authors state that “This study confirms that ALK1 can bind to full-length pro-BMP9 or pro-BMP10 complex without displacing the entire prodomain”. In my opinion this is too speculative and the experiments presented do not really support this. For example, can the observed “weaker” density for the pro-domain in the rather low-resolution pro-BMP9-ALK1 crystal structure be attributed to an only partially occupied molecule? Moreover, for the SPR, can the pro-domain not just be detached from the signalling domain – there is clearly a flexible linker between the pro- and signalling domains? The same could be true for the SEC analysis coupled with SDS-PAGE analysis (Fig. 6).

To understand the real conformation of the pro-BMP9-ALK1 complex in solution, additional experiments could be carried out – for example SAXS, which may be useful to confirm the observed complex in the crystal structure. I suggest to carry out such an analysis, or alternatively, tune down the physiological relevance of the pro-BMP9-ALK1 complex observed in the crystal structure, since this also seems to be a controversial point in the literature.

Minor comments:

1. Wouldn't it be useful to list the genes labelled with red dots in Fig. 1 in an additional supplementary Table?
2. SPR analysis. CM5 chips have 4 flow cells. Was flow cell number 1 run without immobilised ALK1-Fc or ALK1? Was a signal from a control flow cell (1?) subtracted from flow cells with ALK1 during the analysis?
3. SPR analysis. How long (minutes?) and at what speed, was the regeneration with guanidinium chloride performed?
4. Fig. 2A: How was the superposition calculated – based on the BMP9 dimer? It looks to me that the receptors are somehow asymmetric?
5. Fig. 3b. It would be helpful to label all residues represented by spheres.

6. Fig. 4d. Does H73 really form a hydrogen bond with P366? What is the distance between indicated N and O atoms?

7. Fig. 4c-f. It might be helpful to use a darker colour for dashed lines representing hydrogen bonds. Selected distances could be indicated too.

8. Fig. 4d. The side chain of F411 could be shown and might be informative.

9. Fig. 4b-h. Shadows could be switched during rendering in PyMOL to make the figures clearer.

10. Fig. 5c. Please label residues corresponding to blue spheres. Side chains of discussed residues (Y416 in BMP9, F411 in BMP10) and corresponding interactions could be shown in this panel as well.

11. Figs. 5e, 6d, 7a. It would be helpful to label all residues represented with spheres.

12. Fig. 7a. What is coloured in cyan?

13. Page 22, CHAPs should be changed to CHAPS.

14. What volumes of protein and reservoir solutions were used for crystallization?

15. Why were 0 waters built at 2.8 Å but 23 waters at 3.3 Å?

16. What ligands/ions were built in pdb entries 6SF1 and 6SF2?

17. MolProbity clashscores might be informative in Suppl. Table 1.

18. PDB entry 6SF1: How do you know it is Ni and Ca at 2.8 Å? By distances to neighbouring atoms? Similarities to previously published structures? Are these sites conserved in 6SF2 and 6SF3?

19. Fig. 6e. Change y axis label to “Absorbance at 280 nm” instead of “280 nm”.

20. Supplementary Fig. 2. How was the 2FoFc map calculated and at what stage of the refinement process?

21. General comment: For RMSD calculations, it would be useful to specify the residue range used.

Reviewer #3 (Remarks to the Author):

The manuscript entitled ‘Molecular basis of ALK1-mediated signalling by BMP9/BMP10 and their prodomain-bound forms’ authored by Salmon et al. provides a detailed and comprehensive biological and molecular analysis of the endothelial ALK1-mediated BMP signalling pathway focusing on recognition of BMP9/BMP10 by receptors and the consequences for signaling. The BMP9/BMP10 pathway has attracted much attention for its therapeutic potential for treating cardiovascular diseases, such as Pulmonary Arterial Hypertension, and cancers.

Overall, the manuscript is of high technical quality and presents a significant gain in knowledge both with respect to the novelty of data and rectification of several previous findings. As such this manuscript represents an advance in understanding likely to influence thinking in the field.

Minor points:

1) One point of critique is that the manuscript is somewhat unfocused. This is not surprising given the broad scope of the work and the complex underlying biology. A recommendation, therefore, would be to improve the link between sections, perhaps by stating more clearly in the introduction how the different sections fit together. In its current form, the introduction (and to some degree the discussion) reads too much like a list of facts that may or may not be linked by a central hypothesis. The authors should streamline their message around a more central message, as this would make the manuscript more easily accessible to the expected broad audience that will be interested in these findings.

2) A comparison of BMP9 variants (Figure 8) using a smad signaling assay (for example in C2C12 or other cells using p-smad antibodies or other reporters) or, preferably, SPR (as in Figure 1) against the

different receptors would be very instructive in showing whether these variants simply lose activity, or whether they lose the ability to engage a particular receptor that may uniquely control the osteogenic program in C2C12 cells. This evidence would help strengthen the hypothesis that tripartite recognition is a critical feature of receptor binding specificity and for shuttling cells into distinct biological programs. However, although such evidence would be very welcome, it is not essential for publication.

Thank you for reviewing our manuscript and providing constructive critiques. Below is our point-by-point response to the comments, with the original reviewers' comments in *italics* and our response in normal plain text. Changes made in the revised manuscript are highlighted in **red**. Please note that the page and line numbers refer to the marked version of the manuscript.

Reviewers' comments:

Reviewer #1 (Remarks to the Author):

Salmon RM et al investigated the molecular and structural bases of ALK1 in mediating BMP9/BMP10 signaling activities. By analyzing the crystal structures, the authors showed that a tripartite recognition mechanism of BMP10:ALK1 complex defined BMP9 and BMP10 specificity, and that the prodomain-bound BMP9:ALK1 complex represented the circulating BMP9's initial encounter with the high-affinity receptor. Furthermore, structural and experimental evidence did not support Crossveinless 2 as a BMP9 inhibitor. Interestingly, the introduction of BMP10-specific residues into BMP9 abolished BMP9-induced ALP in C2C12 cells, but did not affect BMP9-induced osteogenic activity in vivo. The authors further demonstrated that BMP10 also exhibited osteogenic activity in vivo, which is rather unexpected.

The reported findings are interesting and shed important insights into the role of ALK1 in mediating BMP9 vs. BMP10 signaling. The findings also provide structural evidence about the distinct signaling mechanisms regarding the signal. The reported work represents a significant undertaking. The experimental design was straightforward and reasonably executed. For the most part, data analysis and interpretations were appropriate. While the structural analysis of the BMP-ALK1 interactions was fairly thorough and well carried out, some biological aspects of the studies were not conceived, which might have led to certain unexpected findings. Thus, the manuscript will be significantly improved if the following concerns are fully addressed:

1). 1.1) The commercially available recombinant BMP9 and BMP10 from R & D Systems were used in this study. While these proteins have been used for numerous published studies and may be suitable for protein structure studies, the biological activities of these recombinant proteins may be comprised and are only tested in a limited fashion, i.e., ALP activity in MC-3T3 osteoblastic cells and/or ATDC5 chondrogenic cells. Thus, the validity of the study, especially regarding reported biological (osteogenic) activities of BMP9 vs. BMP10, is a concern. According to the vendor's description, some of the rhBMPs are tagged with 6xHis at C-termini, which may be detrimental to

their biological activities. In fact, this may explain some of the inconsistent observations reported by the authors: rhBMP9 was less effective in inducing osteogenic signaling than that of BMP2; BMP9 with defective ALP activity was still osteogenic in vivo; and rhBMP10 was shown to exhibit osteogenic activity in vivo, which has never been observed in other studies. 1.2) In order to more definitively resolve these unexpected findings, the authors should exogenously express the BMP9 mutants and BMP10 in osteoblast progenitors and verify their osteogenic activities in vitro and/or in vivo.

Response: We have divided this comment into two parts to answer. 1.1) the first part is about the form of the protein used in this manuscript. We totally agree with this reviewer about the limits and disadvantages of the commercially available recombinant BMP9 and BMP10. And for precisely the same reason, we have spent a lot of efforts generating human BMP9 and BMP10 by expressing human full-length cDNA of proBMP9 and proBMP10 in HEK cells without any tags. The recombinant proteins were processed naturally by HEK cells, and purified as prodomain bound complexes, which are the same as the reported *in vivo* circulating forms^{1,2}. We have carried out extensive characterizations of the purified proteins by SDS-PAGE, Biacore binding assays (to ALK1, Fig 1e&f, and to endoglin³), bio-activity assays (signalling assays in PAECs, Fig 1a), ELISA⁴ and X-ray crystallography (this MS).

In Figure 8, apart from BMP2 which is the GF-domain without any tags purchased from R&D Systems, all the remaining BMPs in this figure, including WT BMP9, WT BMP10 and all the BMP9 mutants, are the prodomain bound forms. Apologies that this was not labelled clearly enough in the original figure which might have caused the misunderstanding. We have improved this by i) including a schematic in Supplemental Figure 1 showing the nomenclature of the different forms of BMP9 in this MS; ii) explaining more clearly in the corresponding online methods (page 27, line 602); iii) re-labelling Figure 8 annotation to make this clearer.

We can assure this reviewer that the BMP9 and BMP10 used in Figure 8 are the most physiologically relevant forms currently available in the literature, which are prodomain bound, non-tagged and fully active proteins.

With regard to the comment “*rhBMP10 was shown to exhibit osteogenic activity in vivo, which has never been observed in other studies*”, it is worth pointing out that there is only one report investigating BMP10 osteogenic activity *in vivo*⁵. In this report, the authors have implanted

C2C12 cells, which have been infected with adenovirus containing the full-length proBMP10 cDNA, into the quadriceps muscles of athymic nude mice. Because BMPs are synthesized as pre-proteins and processed by furin during the secretion, it is not known in this model whether proBMP10 produced by the C2C12 cells are fully processed and active.

For the second part of this comment, 1.2) *In order to more definitively resolve these unexpected findings, the authors should exogenously express the BMP9 mutants and BMP10 in osteoblast progenitors and verify their osteogenic activities in vitro and/or in vivo.*

Since BMP9 is mostly produced in the liver ^{1,6,7} and BMP10 is produced from developing hearts and right atria in adult ⁸⁻¹⁰, the osteogenic activity of BMP9 and BMP10 in the muscle will act through paracrine signalling. Hence it is appropriate to use recombinant proteins in the osteogenic assays. In addition, osteoblast progenitors are not the physiological BMP9- and BMP10-producing cells, and they may not be able to process proBMP9 and proBMP10 effectively. Therefore, with respect, we do not feel that exogenously expressing the BMP9 mutants and BMP10 in osteoblast progenitors will provide definitive answers.

2). The biological activities of BMP9 and BMP10 (and their derivatives) were tested in one line, C2C12, which is a myoblastic line and not a prototypic osteoblast progenitor line. Other mesenchymal stem cell (MSC) lines or primary MSCs should be used to validated the findings.

Response: The purpose of this part of the experiment was to show the specificity switch by mutating BMP9 residues to BMP10. To achieve this, we were looking for a well-established assay that shows the difference between BMP9 and BMP10. Kang *et al* ⁵ has nicely demonstrated such difference both in *in vitro* C2C12 assays and in *in vivo* heterotopic bone forming assay, hence we have used ALP assay in C2C12 cells and the *in vivo* HO model.

The same group have also used a second line, C3H10T1/2, to show that BMP9 induced ALP activity whereas BMP10 did not ¹¹. C3H10T1/2 cells are multipotent, mesenchymal stem cell (MSC)-like progenitor cells that are widely used in musculoskeletal research ¹². We have purchased this cell line (C3H/10T1/2, Clone8 (ATCC® CCL-226™)) and performed BMP9 and BMP10-induced ALP activity assays as reported ¹¹. Surprisingly, we can only observe very low ALP activity in this line after 6-9 days of BMP treatment (with the treatment medium replenished

every two days), and crucially, BMP10-induced ALP activity was only marginally lower than BMP9 (Reviewers' Figure 1, two independent experiments). Since the purpose of this set of experiments is to demonstrate the change of ligand specificity from BMP9 to BMP10, this assay in C3H10T1/2 is NOT suitable to address the question. Therefore, we could not add signalling in MSC to address this question. However, as suggested by Reviewer 3, we have performed an additional Smad-based signalling assay in C2C12 cells (Revised Supplemental Figure 9) and an SPR binding assay (Revised Supplemental Figure 10) to further confirm the specificity switch due to mutations, and the results of both assays confirm the change of receptor specificity by replacing BMP9 residues with corresponding ones from BMP10, supporting our model of tripartite recognition.

Reviewers' Figure 1. Signalling assay in C3H10T1/2 cells monitoring induction of ALP activity by BMPs. (a) Cells were seeded in 24 well plate and grown in DMEM containing 0.25% FBS and indicated ligands for 6 days. Media were replenished every 2 days and harvested after 6 days. **(b)** Cells were seeded in 6 well plates and treated with pro-BMP9 or pro-BMP10 at indicated concentrations for 4 or 6 days.

3). As BMP9 and BMP10 can form heterodimers, it's conceivable that the observed BMP10-induced osteogenic activity may be caused by BMP10 "hijacking" BMP9 activity. Thus, BMP9-null or silenced MSCs may be used to determine whether BMP10 is osteogenic per se and/or BMP9-independent.

Response: It is worth noting that only one report so far showing the presence of BMP9/10 heterodimer¹³, and such heterodimer was reactive to antibodies against either BMP10 and

BMP9 in both western and ELISA ¹³. Purified prodomain-bound BMP10 was used in our study, which has identical signalling activity to BMP9 in PAECs (Fig 1a). The purified pro-BMP10 has been confirmed by mass spec peptide mapping, N-terminal sequencing and immunoblotting using BMP10 monoclonal antibodies ². It does not react with BMP9 antibodies either by western blot or by ELISA (sensitivity 9 pg/ml)⁴. Therefore there was no BMP9/10 heterodimer in our assays and the observed BMP10-induced osteogenic activity reported here cannot be due to the heterodimer or BMP10 hijacking BMP9 activity.

4). Along the same line, it's conceivable that BMP10 may exhibit osteogenic activity in osteoblastic differentiation stage-dependent fashion; e.g., higher (or detectable) osteogenicity in more differentiated bone progenitors, but less active in early stage of MSCs. Thus, the authors should again test BMP10 in different bone progenitor cells.

Response: The mechanism of how BMP10 exhibits osteogenic activity is an interesting topic of its own and requires extensive *in vitro* and *in vivo* studies. The focus of the current manuscript is the specificity of ALK1-mediated BMP9 and BMP10 signalling. Therefore, although it is possible, as suggested by this reviewer, that BMP10 may exhibit osteogenic activity in osteoblastic differentiation stage-dependent fashion, it is not the focus of the current manuscript. With respect, we think adding such a body of data will not alter or enhance the conclusion of the paper and may distract the readers from the main message of this manuscript.

5). The authors found CV2 did not inhibit BMP9 signaling in C2C12 cells. Based on the structural analysis, can the authors predict how effectively BMP antagonists such as noggin and gremlin would inhibit BMP9 or BMP10?

Response: The reason we choose to investigate CV2 was because CV2 is the only ligand trap that has been reported to inhibit BMP9 signalling ¹⁴.

Following your suggestion, we have performed structural analysis on Noggin and Gremlin. By overlaying BMP9 structure onto BMP7 in BMP7:Noggin complex, the extra residue in BMP9 and BMP10 (BMP9-Y416 or BMP10-F411) lies right in the middle of the Noggin interaction surface and will have steric hindrance for Noggin binding (Reviewers' Figure 2). Therefore the structure analysis would predict Noggin cannot bind to BMP9 or BMP10 in a similar manner as

it binding to BMP7. This is in agreement with the published experimental result that Noggin does not inhibit BMP9 signalling activity¹⁵.

Overlaying BMP9 with Grem2:GDF5 structure shows the insertion in BMP9 and BMP10 does not have a direct impact on Gremlin interaction with BMPs, hence no direct conclusion can be obtained from the structural analysis. We will test experimentally whether Gremlin inhibits BMP9 or BMP10 signalling in a follow-up study. As this is not the focus of the current manuscript, we will not include such data in the revised manuscript.

Thank you for raising this interesting point.

Reviewers' Figure 2. Structural analysis of Noggin effects on BMP9. BMP9 (pdb code: 4mpl, in magenta) is overlaid onto BMP7:Noggin structure (pdb code: 1M4U, Noggin in green and BMP7 in cyan), with an overview in (a) and zoomed-in views in (b). Noggin does not interact with BMP7 through extended β -sheet, instead, it is like a saddle sitting on top of BMP7, covering both the type I and the type II receptor binding sites. The insertion in BMP9 and BMP10 (only BMP9 F416 is shown here) locates right at the centre where Noggin interacts with BMP7, and when overlaid, F416 clashes with Noggin. Therefore the prediction from the structural analysis is that Noggin will not inhibit BMP9 (or BMP10) signalling, which has been tested and confirmed experimentally by David et al¹⁵.

6). In Figure 8, it was not clear why Cardiotoxin was used to stimulate *in vivo* HO. The retrieved HO bony masses should be examined by H & E and trichrome staining to verify heterotopic ossification.

Response: It has been reported that BMP9-induced muscle heterotopic ossification requires

changes to the skeletal muscle microenvironment and cardiotoxin can enhance the effect of HO by recombinant BMP9¹⁶. We need a robust model of BMP9-induced *in vivo* ossification to evaluate whether any mutation can modify this effect, hence we performed the *in vivo* HO experiment in the presence and absence of cardiotoxin pre-treatment. We found as reported, cardiotoxin-enhanced heterotopic bone formation. **We have added a sentence with reference to the above paper in the online methods section (page 28, line 610)** to make this point clearer.

In addition, we have examined the retrieved HO bony mass by H&E and trichrome staining as suggested, and **included in the revised manuscript (Figure 8c, page 28, line 616)**.

7). *Minor comments: PAECs and PSMCs were not defined in the text in their first appearance.*

Response: Thank you! We have now corrected this in the revised manuscript.

Reviewer #2 (Remarks to the Author):

The authors illuminated the molecular mechanisms of BMP9/BMP10-ALK1 signaling using a wide range of methods including crystallography, biophysical binding assays, cellular and in vivo mouse studies. They reveal a rather unexpected role of the pro-domains of BMP9/10 and their passive role in signalling and binding to ALK1. The authors have performed detailed structural analyses of BMP10-ALK1 and pro-BMP9-ALK1 complexes. These structural analyses then shed light on our understanding of the role of CV2 and guided site-directed mutagenesis of BMP9/10, cellular and in vivo assays to decipher BMP9/10 specificity. The results are of high quality, timely and of high interest to a wide community of structural and developmental biologists, cancer researchers and drug discoverers due to a wide range of mechanisms controlled by BMP9/10 in human health and disease.

Response: Thank you for your positive comments. Really appreciated.

However, I have one major concern, which would need addressing - the co-existence of the pro-BMP9-ALK1 complex. The authors state that "This study confirms that ALK1 can bind to full-length pro-BMP9 or pro-BMP10 complex without displacing the entire prodomain". In my opinion this is too speculative and the experiments presented do not really support this. For example, can the observed "weaker" density for the pro-domain in the rather low-resolution pro-BMP9-ALK1 crystal structure be attributed to an only partially occupied molecule?

Moreover, for the SPR, can the pro-domain not just be detached from the signalling domain – there is clearly a flexible linker between the pro- and signalling domains? The same could be true for the SEC analysis coupled with SDS-PAGE analysis (Fig. 6).

To understand the real conformation of the pro-BMP9-ALK1 complex in solution, additional experiments could be carried out – for example SAXS, which may be useful to confirm the observed complex in the crystal structure. I suggest to carry out such an analysis, or alternatively, tune down the physiological relevance of the pro-BMP9-ALK1 complex observed in the crystal structure, since this also seems to be a controversial point in the literature.

Response: Thank you for your comments and suggestions. We have addressed your concern from the following three aspects: 1) clarifying the current data; 2) performing a further experiment to provide additional evidence of pro-BMP9-ALK1 ternary complex; 3) tuning down the physiological relevance as suggested.

Clarifying the current data

1) *“can the observed “weaker” density for the pro-domain in the rather low-resolution pro-BMP9-ALK1 crystal structure be attributed to an only partially occupied molecule”.*

We understand that the reviewer has to make a judgement without access to the pdb file which is difficult. We can assure this reviewer that the density at the BMP9:prodomain interface is very clear, B factors are in the similar range to the BMP10 residues in the same region (**revised Supplemental Figure 6a**), and we can build all the residues with full occupancy. To assist your evaluation, we have now **updated the supplemental Figure 6 to include both the B factor and the density in the prodomain:GF domain interface**, and included sections of the pdb coordinate file for the interface residues in Appendix 1 of this response to the reviewer’s comments, which contains the actual numbers for the occupancy and B factors for each atom in this area.

2) *“Moreover, for the SPR, can the pro-domain not just be detached from the signalling domain – there is clearly a flexible linker between the pro- and signalling domains? The same could be true for the SEC analysis coupled with SDS-PAGE analysis (Fig. 6).”*

For all the pro-BMP9 and pro-BMP10 samples used in this manuscript, including SPR, SEC, native PAGE and X-ray crystallographic studies, prodomains are non-covalently bound to the GF-domains (signalling domain), therefore if detached, it will no longer be linked to the GF-domain and should be separated from the GF-domain. Maybe this was not stated clearly enough in our original manuscript and abbreviation list. We have now: i) introduced **Supplemental Figure 1** to show the schematic of BMP9 processing and the nomenclatures. ii) **modified the text and explained it explicitly in the abbreviation and the first appearance** to emphasize that we are using the non-covalently linked complex in this manuscript.

We only have one set of SPR data in Figure 1e&f; we assume the reviewer refers to this data. The aim of this data is to evaluate binding affinities. Due to the steric hindrance of ALK1 on the Biacore chip, we do not think this is a good system to evaluate whether the prodomain is detached from the GF-domain, since it does not reflect the solution or the cell surface

scenario. Therefore, we did not use this experiment to support the pro-BMP9:ALK1 complex formation.

Additional evidence supporting prodomain-BMP9:ALK1 complex:

- 1) As ALK1 ECD is very small (~10kD) and binds to pro-BMP9 at the wrist epitope. The shape changes between pro-BMP9 and pro-BMP9:ALK1 complex will be very small and may be difficult to distinguish by SAXS. Here we presented the X-ray crystal structure at 3.3 Å resolution and as described above, the interface between prodomain and GF-domain in our structure is well defined. Therefore we do not feel SAXS will give us more definitive answer than the current crystal structure which has much higher resolution.
- 2) We have performed an additional solution experiment with more controls to strengthen our data. For the SEC analysis, if the prodomain (37kDa) is detached from the signalling domain, it should elute at a much later position on the SEC column. In the original submission we did not have such a control to demonstrate this. We have recently shown³ that soluble endoglin (sENG) can displace the prodomain from pro-BMP9 complex. In the revised manuscript, we have performed sENG displacement experiment as a positive control alongside pro-BMP9 + ALK1 run under the identical condition (**revised Figure 6e-h. The Figure 6e&f in the original submission has now been moved to Supplemental Figure 8 in the revised manuscript**) and confirmed that while displaced prodomain by sENG can be detected as a new peak in the SEC run, there is no free prodomain released upon mixing pro-BMP9 with ALK1 ECD in solution, and the prodomain remains in the same fractions as ALK1 and BMP9, confirming the presence of the pro-BMP9:ALK1 complex.

Tuning down the physiological relevance

We appreciate the complexity of cell surface receptors and co-receptors distribution and their influence on the initial encounter complex at the cell surface, we have tuned down the physiological relevance as suggested by this reviewer by: **i) removing the statement “a novel snapshot of a prodomain-bound ligand complex contacting a cell surface receptor” in the abstract (Page 3) and the discussion (last paragraph on page 19). ii) removing one paragraph in the discussion (page 18).**

Minor comments:

1. Wouldn't it be useful to list the genes labelled with red dots in Fig. 1 in an additional supplementary Table?

Response: Thank you for this suggestion. They are now included in Supplemental Dataset S1 and Dataset S2.

2. SPR analysis. CM5 chips have 4 flow cells. Was flow cell number 1 run without immobilised ALK1-Fc or ALK1? Was a signal from a control flow cell (1?) subtracted from flow cells with ALK1 during the analysis?

Response: Yes to both questions. Apologies we did not specify this in the initial submission. We have added such information in the online methods section (page 23 line 498) in the revised manuscript.

3. SPR analysis. How long (minutes?) and at what speed, was the regeneration with guanidinium chloride performed?

Response: The regeneration was performed with four times, 8 sec injection of 4M Guanidine hydrochloride at a flow rate of 30 μ l/min. A surface performance test was run after the chip was generated to ensure the traces from two cycle with the identical ligand concentration overlap well. This information is now added in the online methods section (page 23 line 503).

4. Fig. 2A: How was the superposition calculated – based on the BMP9 dimer? It looks to me that the receptors are somehow asymmetric?

Response: The superposition was calculated using PyMOL, based on BMP10 dimer. For BMP10:ALK1 structure, there is only one copy of BMP10 and ALK1 monomer in an asymmetric unit, so the dimeric unit of BMP10 dimer:ALK1 is symmetrical. However, in the ALK1:BMP9:ActRIIb structure (pdb code: 4FAO), there are 6 copies of ternary complexes in an asymmetric unit (12 chains of BMP9, 12 chains of ALK1 and 12 chains of ActRIIb). Small changes can be observed in the ALK1 loops among the 6 complex copies (Reviewers' Figure 3a). The original publication for the ternary complex of 4FAO did not specify which complex was used in the analysis¹⁷, therefore we have used one of the complex in 4FAO containing chains M,

N, Q, R, O and P for all the analysis in our manuscript. Upon close examination, the two monomers of BMP9 and ALK1 in this complex are not symmetrical in some ALK1 loops (Reviewers' Figure 3b), so the apparent asymmetrical look may arise from a combination of asymmetry in the 4FAO model and the angle of the display. Since this difference is not the focus in our manuscript and would not alter the conclusion of the paper, we have generated a new version of the figure to minimise the asymmetrical effect (**revised Figure 2a**).

Reviewers' Figure 3. Comparison of ALK1:BMP9 units in 4FAO. There are a total of 36 peptide chains, six copies of ALK1:BMP9:ActRIIb ternary complexes in one asymmetric unit of 4FAO. **(a)** Overlay of the six copies of the complexes in 4FAO, showing that ALK1 finger 2 loops have different conformations among different copies. **(b)** When the two monomers of BMP9:ALK1 in the complex used in the paper are overlaid, the ALK1 loops display different conformation, suggesting the two protomers of ALK1 in this complex are not in exactly the same conformation. This partially explains why Figure 2a does not look symmetrical.

5. Fig. 3b. It would be helpful to label all residues represented by spheres.

Response: This is a good idea - thank you for the suggestion. We found that adding labels on the original Figure 3B makes it very crowded, as it contains the overlay of 4 BMPs. We therefore included a new panel (**revised Figure 3b**) which displays all the residues represented by spheres on BMP10 structure, and **change the original Figure 3b to Figure 3c**.

6. Fig. 4d. Does H73 really form a hydrogen bond with P366? What is the distance between indicated N and O atoms?

Response: 3.7Å. Yes we think so.

7. Fig. 4c-f. It might be helpful to use a darker colour for dashed lines representing hydrogen bonds. Selected distances could be indicated too.

Response: Thank you for your suggestion. We have changed the bond colour from yellow to orange. Also we have turned on the label function in PyMOL so all the bond lengths are now labelled.

8. Fig. 4d. The side chain of F411 could be shown and might be informative.

Response: We have added the side chain of F411 as suggested.

9. Fig. 4b-h. Shadows could be switched during rendering in PyMOL to make the figures clearer.

Response: We have generated new versions of Figure 4b-h and turned off shadows during rendering. Thank you for the suggestion.

10. Fig. 5c. Please label residues corresponding to blue spheres. Side chains of discussed residues (Y416 in BMP9, F411 in BMP10) and corresponding interactions could be shown in this panel as well.

Response: Thank you. We have labelled all the blue spheres and showed side chains as suggested in the revised manuscript.

11. Figs. 5e, 6d, 7a. It would be helpful to label all residues represented with spheres.

Response: We have labelled all the residues represented with spheres.

12. Fig. 7a. What is coloured in cyan?

Response: Cyan is the other monomer of BMP2. We realised this was confusing, therefore **in the revised version, we have coloured both monomers of BMP2 in green.** Thank you for pointing this out.

13. Page 22, CHAPs should be changed to CHAPS.

Response: Thank you! **We have corrected this.**

14. What volumes of protein and reservoir solutions were used for crystallization?

Response: We used the hanging drop setup using the pre-greased 24 well plates from Hampton Research, 500 µl of reservoir solution, 1 µl protein + 1 µl reservoir solution for the drop. **We have now added this information in the materials and methods (page 24, line 521).**

15. Why were 0 waters built at 2.8 Å but 23 waters at 3.3 Å?

Response: We have gone through the two structures again and refined the water molecules in both structures. Five water molecules have now been built in the 6SF1 structure (2.8 Å) and 9 water molecules in the 6SF2 (3.3 Å) structure. The original difference might be because the two structures were solved one year apart and by different operators. **We have re-deposited the newly refined structures and the updated pdb validation reports are attached.**

16. What ligands/ions were built in pdb entries 6SF1 and 6SF2?

Response: Two NAG (N-Acetylglucosamine) molecules were built in 6SF2 which are the N-linked glycosylation on residue Asn136 in chain C and F. Five nickel ions have been built in 6SF1.

17. MolProbity clashscores might be informative in Suppl. Table 1.

Response: **We have now added this information in Suppl. Table 1.**

18. PDB entry 6SF1: How do you know it is Ni and Ca at 2.8 Å? By distances to neighbouring atoms? Similarities to previously published structures? Are these sites conserved in 6SF2 and 6SF3?

Response: Thank you for pointing this out. We have checked all the structures again. For 6SF1, we refined the blob difference density maps against all the possible ion components found in the purification buffers and crystallization conditions, including Ni²⁺, Na⁺, Cl⁻, I⁻ and water molecules. Only nickel ions were refined well and they are well-coordinated by the surrounding atoms although these atoms were not close enough to form very strong interactions (Reviewers' Figure 4). No Ca in the condition and **this is corrected in the new pdb file**. These sites are not conserved in 6SF2 or 6SF3, hence it is likely due to the specific crystallization condition, and they are not the discussion points in the manuscript. All the BMP10:ALK1 structural analysis was carried out using 6SF3, which is a higher resolution version of 6SF1. Because we obtained 6SF1 structure more than one year before we collected 6SF3 dataset, we have used 6SF1 coordinates to do the molecular replacement. In addition, 6SF1 and 6SF3 were crystallized under slightly different conditions, hence, we have reported both structures in this manuscript.

Reviewers' Figure 4. Comparison of fitting water or Nickel into the five density blobs. The 2Fo-Fc (1.5 σ) and Fo-Fc (3.0 σ) maps of the nickel binding sites in 6SF1 generated using Coot from the final stage of refinement. These sites were refined with all possible ions (components of the buffers including Ni²⁺, Na⁺, Cl⁻, I⁻) and water molecules. Only Ni²⁺ refined well and they coordinate well with the surrounding atoms although they are not close enough to form very strong interaction.

19. *Fig. 6e. Change y axis label to “Absorbance at 280 nm” instead of “280 nm”.*

Response: Thank you. **We have now corrected this in the revision.**

20. *Supplementary Fig. 2. How was the 2FoFc map calculated and at what stage of the refinement process?*

Response: The 2Fo-Fc map was calculated by Phenix.refine in the final round of the refinement and output as .mtz file. The mtz file was then converted to the .map file using ‘Create a map from map coefficients’ programme in the PHENIX suite and displayed using PyMOL.

21. *General comment: For RMSD calculations, it would be useful to specify the residue range used.*

Response: Thank you. We have corrected this at 1) **Figure 2a and page 8 line 155 of the text;** and 2) **Figure 6b, and page 12, line 270 of the text.**

Reviewer #3 (Remarks to the Author):

The manuscript entitled 'Molecular basis of ALK1-mediated signalling by BMP9/BMP10 and their prodomain-bound forms' authored by Salmon et al. provides a detailed and comprehensive biological and molecular analysis of the endothelial ALK1-mediated BMP signalling pathway focusing on recognition of BMP9/BMP10 by receptors and the consequences for signaling. The BMP9/BMP10 pathway has attracted much attention for its therapeutic potential for treating cardiovascular diseases, such as Pulmonary Arterial Hypertension, and cancers.

Overall, the manuscript is of high technical quality and presents a significant gain in knowledge both with respect to the novelty of data and rectification of several previous findings. As such this manuscript represents an advance in understanding likely to influence thinking in the field.

Response: Thank you for your very kind comments.

Minor points:

1) One point of critique is that the manuscript is somewhat unfocused. This is not surprising given the broad scope of the work and the complex underlying biology. A recommendation, therefore, would be to improve the link between sections, perhaps by stating more clearly in the introduction how the different sections fit together. In its current form, the introduction (and to some degree the discussion) reads too much like a list of facts that may or may not be linked by a central hypothesis. The authors should streamline their message around a more central message, as this would make the manuscript more easily accessible to the expected broad audience that will be interested in these findings.

Response: Thank you very much for the constructive comments. In the revised manuscript, we have incorporated your suggestions and modified the manuscript at the following places: **abstract, last paragraph of the introduction (page 6), discussion on page 18 and 19.** Now the flow of the manuscript follows the link of two novel crystal structures and the analysis/validation of the structural findings, which brings the central message of a tripartite recognition mechanism that defines the specificity of BMP9 and BMP10 for ALK1-mediated signalling.

2) A comparison of BMP9 variants (Figure 8) using a smad signaling assay (for example in C2C12

or other cells using p-smad antibodies or other reporters) or, preferably, SPR (as in Figure 1) against the different receptors would be very instructive in showing whether these variants simply lose activity, or whether they lose the ability to engage a particular receptor that may uniquely control the osteogenic program in C2C12 cells. This evidence would help strengthen the hypothesis that tripartite recognition is a critical feature of receptor binding specificity and for shuttling cells into distinct biological programs. However, although such evidence would be very welcome, it is not essential for publication.

Response: Thank you for your suggestion. 1) those BMP9 variants that lost activity in C2C12 cells are not because the mutants simply lost activity, as they are all fully active when signalling in PAECs (Figure 8a, white bars). 2) We agree with you that using a Smad signalling assay or SPR would be very instructive in showing receptor specificity. We therefore performed a Smad-mediated signalling assay (using *ID1* gene induction after 1-hour treatment as a readout) (Revised Supplemental Figure 9) as well as additional SPR experiments (Revised Supplemental Figure 10) as suggested. In the C2C12-ID1 gene induction signalling assays, because the ligands are predicted to use a similar set of receptors to the ALP assays, we would expect to see a similar pattern of activity as in the ALP assay amongst the BMP9 mutants, for example, E337K would have WT BMP9-like activity and D366E would not have any signalling activity. Indeed, we saw such a result. In the SPR experiment, we used D366E as an example (Revised Supplemental Figure 10), and showed that pro-BMP9, pro-BMP10 and pro-BMP9 D366E all bind ALK1-Fc within 10-100 picomolar range, but they have very different binding affinities for ALK2-Fc. While BMP9, whether the prodomain bound form or the GF-domain alone, binds ALK2-Fc with K_D value of around 85 nM, we cannot detect pro-BMP10 binding to ALK2-Fc by SPR. Similarly, BMP9 D366E mutant which lost ALP activity in C2C12 cells also lost binding to ALK2. It has been reported previously that BMP9 signalling in endothelial cells is fully mediated by ALK1¹⁸, whereas both ALK1 and ALK2 are required for BMP9 signalling in C2C12 cells¹⁹, therefore the SPR data support our hypothesis that mutating BMP9 residue D366 into BMP10 residue Glu has converted BMP9 which is capable of binding to ALK2, into a BMP10-like ligand which does not bind ALK2.

Reference

- 1 Bidart, M. *et al.* BMP9 is produced by hepatocytes and circulates mainly in an active mature form complexed to its prodomain. *Cell. Mol. Life Sci.* **69**, 313-324, doi:10.1007/s00018-011-0751-1 (2012).
- 2 Jiang, H. *et al.* The Prodomain-bound Form of Bone Morphogenetic Protein 10 Is Biologically Active on Endothelial Cells. *J. Biol. Chem.* **291**, 2954-2966, doi:10.1074/jbc.M115.683292 (2016).
- 3 Lawera, A. *et al.* Role of soluble endoglin in BMP9 signaling. *Proc. Natl. Acad. Sci. U. S. A.*, doi:10.1073/pnas.1816661116 (2019).
- 4 Hodgson, J. *et al.* Characterization of GDF2 Mutations and Levels of BMP9 and BMP10 in Pulmonary Arterial Hypertension. *Am. J. Respir. Crit. Care Med.*, doi:10.1164/rccm.201906-1141OC (2019).
- 5 Kang, Q. *et al.* Characterization of the distinct orthotopic bone-forming activity of 14 BMPs using recombinant adenovirus-mediated gene delivery. *Gene Ther.* **11**, 1312-1320, doi:10.1038/sj.gt.3302298 (2004).
- 6 Miller, A. F., Harvey, S. A., Thies, R. S. & Olson, M. S. Bone morphogenetic protein-9. An autocrine/paracrine cytokine in the liver. *J. Biol. Chem.* **275**, 17937-17945 (2000).
- 7 Breilkopf-Heinlein, K. *et al.* BMP-9 interferes with liver regeneration and promotes liver fibrosis. *Gut* **66**, 939-954, doi:10.1136/gutjnl-2016-313314 (2017).
- 8 Neuhaus, H., Rosen, V. & Thies, R. S. Heart specific expression of mouse BMP-10 a novel member of the TGF-beta superfamily. *Mech. Dev.* **80**, 181-184 (1999).
- 9 Somi, S., Buffing, A. A., Moorman, A. F. & Van Den Hoff, M. J. Expression of bone morphogenetic protein-10 mRNA during chicken heart development. *Anat. Rec. A Discov. Mol. Cell. Evol. Biol.* **279**, 579-582, doi:10.1002/ar.a.20052 (2004).
- 10 Chen, H. *et al.* BMP10 is essential for maintaining cardiac growth during murine cardiogenesis. *Development* **131**, 2219-2231, doi:10.1242/dev.01094 (2004).
- 11 Cheng, H. *et al.* Osteogenic activity of the fourteen types of human bone morphogenetic proteins (BMPs). *J. Bone Joint Surg. Am.* **85**, 1544-1552, doi:10.2106/00004623-200308000-00017 (2003).
- 12 Ker, D. F., Sharma, R., Wang, E. T. & Yang, Y. P. Development of mRuby2-Transfected C3H10T1/2 Fibroblasts for Musculoskeletal Tissue Engineering. *PLoS One* **10**, e0139054, doi:10.1371/journal.pone.0139054 (2015).
- 13 Tillet, E. *et al.* A heterodimer formed by bone morphogenetic protein 9 (BMP9) and BMP10 provides most BMP biological activity in plasma. *J. Biol. Chem.* **293**, 10963-10974, doi:10.1074/jbc.RA118.002968 (2018).
- 14 Yao, Y. *et al.* Crossveinless 2 regulates bone morphogenetic protein 9 in human and mouse vascular endothelium. *Blood* **119**, 5037-5047, doi:10.1182/blood-2011-10-385906 (2012).
- 15 David, L. *et al.* Bone morphogenetic protein-9 is a circulating vascular quiescence factor. *Circ. Res.* **102**, 914-922, doi:10.1161/CIRCRESAHA.107.165530 (2008).
- 16 Leblanc, E. *et al.* BMP-9-induced muscle heterotopic ossification requires changes to the skeletal muscle microenvironment. *J. Bone Miner. Res.* **26**, 1166-1177, doi:10.1002/jbmr.311 (2011).
- 17 Townson, S. A. *et al.* Specificity and structure of a high affinity activin receptor-like kinase 1 (ALK1) signaling complex. *J. Biol. Chem.* **287**, 27313-27325, doi:10.1074/jbc.M112.377960 (2012).

- 18 Scharpfenecker, M. *et al.* BMP-9 signals via ALK1 and inhibits bFGF-induced endothelial cell proliferation and VEGF-stimulated angiogenesis. *J. Cell Sci.* **120**, 964-972, doi:10.1242/jcs.002949 (2007).
- 19 Luo, J. *et al.* TGFbeta/BMP type I receptors ALK1 and ALK2 are essential for BMP9-induced osteogenic signaling in mesenchymal stem cells. *J. Biol. Chem.* **285**, 29588-29598, doi:10.1074/jbc.M110.130518 (2010).

Appendix 1: Partial pdb coordinate for 6SF2, pro:BMP9:ALK1 structure, containing the prodomain (chain F) and BMP9 GF-domain (chain E) interface residues. Only the residues at the prodomain:GFD interface in Supplemental Figure 6a are shown.

ATOM	3227	N	GLN	F	87	-20.647	-7.190	17.369	1.00	84.12	BZ00	N
ATOM	3228	CA	GLN	F	87	-19.673	-8.274	17.427	1.00	80.81	BZ00	C
ATOM	3229	C	GLN	F	87	-20.334	-9.644	17.397	1.00	78.66	BZ00	C
ATOM	3230	O	GLN	F	87	-19.682	-10.627	17.028	1.00	86.27	BZ00	O
ATOM	3231	CB	GLN	F	87	-18.812	-8.142	18.684	1.00	85.05	BZ00	C
ATOM	3232	CG	GLN	F	87	-17.651	-7.174	18.550	1.00	90.30	BZ00	C
ATOM	3233	CD	GLN	F	87	-16.485	-7.774	17.792	1.00	100.03	BZ00	C
ATOM	3234	OE1	GLN	F	87	-16.438	-8.983	17.561	1.00	102.53	BZ00	O
ATOM	3235	NE2	GLN	F	87	-15.534	-6.932	17.402	1.00	105.44	BZ00	N
ATOM	3236	N	TYR	F	88	-21.609	-9.731	17.779	1.00	74.84	BZ00	N
ATOM	3237	CA	TYR	F	88	-22.312	-11.008	17.737	1.00	75.61	BZ00	C
ATOM	3238	C	TYR	F	88	-22.423	-11.525	16.308	1.00	79.30	BZ00	C
ATOM	3239	O	TYR	F	88	-22.307	-12.733	16.065	1.00	83.38	BZ00	O
ATOM	3240	CB	TYR	F	88	-23.694	-10.851	18.370	1.00	75.16	BZ00	C
ATOM	3241	CG	TYR	F	88	-24.490	-12.130	18.485	1.00	75.40	BZ00	C
ATOM	3242	CD1	TYR	F	88	-24.220	-13.051	19.488	1.00	78.68	BZ00	C
ATOM	3243	CD2	TYR	F	88	-25.526	-12.405	17.605	1.00	80.09	BZ00	C
ATOM	3244	CE1	TYR	F	88	-24.953	-14.216	19.603	1.00	85.32	BZ00	C
ATOM	3245	CE2	TYR	F	88	-26.264	-13.567	17.711	1.00	85.62	BZ00	C
ATOM	3246	CZ	TYR	F	88	-25.974	-14.469	18.712	1.00	86.51	BZ00	C
ATOM	3247	OH	TYR	F	88	-26.708	-15.628	18.820	1.00	90.10	BZ00	O
ATOM	3248	N	MET	F	89	-22.638	-10.624	15.347	1.00	75.03	BZ00	N
ATOM	3249	CA	MET	F	89	-22.720	-11.039	13.950	1.00	69.34	BZ00	C
ATOM	3250	C	MET	F	89	-21.366	-11.509	13.433	1.00	74.29	BZ00	C
ATOM	3251	O	MET	F	89	-21.291	-12.474	12.662	1.00	79.38	BZ00	O
ATOM	3252	CB	MET	F	89	-23.254	-9.891	13.094	1.00	61.38	BZ00	C
ATOM	3253	CG	MET	F	89	-24.679	-9.480	13.425	1.00	59.15	BZ00	C
ATOM	3254	SD	MET	F	89	-25.906	-10.703	12.921	1.00	63.35	BZ00	S
ATOM	3255	CE	MET	F	89	-25.844	-10.538	11.136	1.00	62.33	BZ00	C
ATOM	3256	N	ILE	F	90	-20.286	-10.842	13.846	1.00	72.30	BZ00	N
ATOM	3257	CA	ILE	F	90	-18.952	-11.248	13.414	1.00	69.17	BZ00	C
ATOM	3258	C	ILE	F	90	-18.598	-12.612	13.992	1.00	72.99	BZ00	C
ATOM	3259	O	ILE	F	90	-18.131	-13.507	13.276	1.00	76.44	BZ00	O
ATOM	3260	CB	ILE	F	90	-17.913	-10.184	13.807	1.00	62.62	BZ00	C
ATOM	3261	CG1	ILE	F	90	-18.196	-8.869	13.081	1.00	64.29	BZ00	C
ATOM	3262	CG2	ILE	F	90	-16.510	-10.675	13.496	1.00	58.98	BZ00	C
ATOM	3263	CD1	ILE	F	90	-17.190	-7.779	13.378	1.00	65.88	BZ00	C
ATOM	3264	N	ASP	F	91	-18.810	-12.788	15.298	1.00	76.91	BZ00	N
ATOM	3265	CA	ASP	F	91	-18.557	-14.084	15.919	1.00	75.87	BZ00	C
ATOM	3266	C	ASP	F	91	-19.415	-15.168	15.285	1.00	63.81	BZ00	C
ATOM	3267	O	ASP	F	91	-18.965	-16.306	15.109	1.00	63.44	BZ00	O
ATOM	3268	CB	ASP	F	91	-18.815	-14.004	17.423	1.00	86.51	BZ00	C
ATOM	3269	CG	ASP	F	91	-17.997	-12.921	18.095	1.00	96.04	BZ00	C
ATOM	3270	OD1	ASP	F	91	-17.185	-12.274	17.401	1.00	98.41	BZ00	O
ATOM	3271	OD2	ASP	F	91	-18.170	-12.714	19.315	1.00	100.09	BZ00	O1-
ATOM	3272	N	LEU	F	92	-20.654	-14.831	14.927	1.00	57.50	BZ00	N
ATOM	3273	CA	LEU	F	92	-21.513	-15.792	14.247	1.00	66.63	BZ00	C
ATOM	3274	C	LEU	F	92	-20.954	-16.146	12.872	1.00	72.83	BZ00	C
ATOM	3275	O	LEU	F	92	-21.060	-17.296	12.427	1.00	70.78	BZ00	O
ATOM	3276	CB	LEU	F	92	-22.930	-15.227	14.141	1.00	71.12	BZ00	C
ATOM	3277	CG	LEU	F	92	-24.105	-16.204	14.150	1.00	71.70	BZ00	C
ATOM	3278	CD1	LEU	F	92	-25.370	-15.485	14.585	1.00	56.45	BZ00	C
ATOM	3279	CD2	LEU	F	92	-24.294	-16.823	12.780	1.00	74.24	BZ00	C
ATOM	3280	N	TYR	F	93	-20.341	-15.172	12.194	1.00	71.46	BZ00	N
ATOM	3281	CA	TYR	F	93	-19.699	-15.442	10.911	1.00	69.68	BZ00	C
ATOM	3282	C	TYR	F	93	-18.515	-16.388	11.076	1.00	72.58	BZ00	C
ATOM	3283	O	TYR	F	93	-18.387	-17.373	10.338	1.00	67.77	BZ00	O
ATOM	3284	CB	TYR	F	93	-19.257	-14.127	10.265	1.00	74.92	BZ00	C
ATOM	3285	CG	TYR	F	93	-18.240	-14.289	9.155	1.00	82.15	BZ00	C
ATOM	3286	CD1	TYR	F	93	-18.624	-14.714	7.889	1.00	84.45	BZ00	C
ATOM	3287	CD2	TYR	F	93	-16.897	-14.005	9.371	1.00	83.89	BZ00	C
ATOM	3288	CE1	TYR	F	93	-17.698	-14.861	6.872	1.00	84.68	BZ00	C
ATOM	3289	CE2	TYR	F	93	-15.964	-14.149	8.360	1.00	87.82	BZ00	C
ATOM	3290	CZ	TYR	F	93	-16.369	-14.577	7.112	1.00	86.67	BZ00	C
ATOM	3291	OH	TYR	F	93	-15.442	-14.720	6.104	1.00	82.08	BZ00	O
ATOM	3292	N	ASN	F	94	-17.636	-16.103	12.043	1.00	77.59	BZ00	N
ATOM	3293	CA	ASN	F	94	-16.498	-16.984	12.292	1.00	75.74	BZ00	C
ATOM	3294	C	ASN	F	94	-16.946	-18.374	12.719	1.00	79.16	BZ00	C
ATOM	3295	O	ASN	F	94	-16.261	-19.362	12.430	1.00	80.50	BZ00	O
ATOM	3296	CB	ASN	F	94	-15.584	-16.377	13.354	1.00	75.41	BZ00	C
ATOM	3297	CG	ASN	F	94	-15.050	-15.022	12.951	1.00	83.74	BZ00	C
ATOM	3298	OD1	ASN	F	94	-14.927	-14.721	11.765	1.00	89.55	BZ00	O
ATOM	3299	ND2	ASN	F	94	-14.728	-14.194	13.938	1.00	87.67	BZ00	N
ATOM	3300	N	ARG	F	95	-18.081	-18.474	13.412	1.00	83.28	BZ00	N
ATOM	3301	CA	ARG	F	95	-18.629	-19.787	13.727	1.00	88.92	BZ00	C
ATOM	3302	C	ARG	F	95	-19.178	-20.469	12.481	1.00	97.25	BZ00	C
ATOM	3303	O	ARG	F	95	-19.152	-21.702	12.389	1.00	99.58	BZ00	O
ATOM	3304	CB	ARG	F	95	-19.717	-19.665	14.794	1.00	85.85	BZ00	C
ATOM	3305	N	TYR	F	96	-19.672	-19.689	11.515	1.00	101.34	BZ00	N
ATOM	3306	CA	TYR	F	96	-20.155	-20.275	10.268	1.00	102.27	BZ00	C
ATOM	3307	C	TYR	F	96	-19.007	-20.806	9.419	1.00	100.72	BZ00	C
ATOM	3308	O	TYR	F	96	-19.134	-21.861	8.786	1.00	100.20	BZ00	O
ATOM	3309	CB	TYR	F	96	-20.963	-19.247	9.475	1.00	104.89	BZ00	C
ATOM	3310	CG	TYR	F	96	-22.447	-19.243	9.766	1.00	107.46	BZ00	C

ATOM	3311	CD1	TYR	F	96	-22.961	-19.877	10.890	1.00110.97	BZ00	C
ATOM	3312	CD2	TYR	F	96	-23.336	-18.614	8.905	1.00105.60	BZ00	C
ATOM	3313	CE1	TYR	F	96	-24.321	-19.874	11.151	1.00110.53	BZ00	C
ATOM	3314	CE2	TYR	F	96	-24.693	-18.607	9.155	1.00106.38	BZ00	C
ATOM	3315	CZ	TYR	F	96	-25.181	-19.237	10.278	1.00108.47	BZ00	C
ATOM	3316	OH	TYR	F	96	-26.535	-19.226	10.521	1.00107.54	BZ00	O

ATOM	3376	N	ILE	F	108	-30.181	-17.234	6.183	1.00 80.61	BZ00	N
ATOM	3377	CA	ILE	F	108	-30.007	-15.789	6.260	1.00 78.90	BZ00	C
ATOM	3378	C	ILE	F	108	-30.125	-15.353	7.713	1.00 74.94	BZ00	C
ATOM	3379	O	ILE	F	108	-30.992	-15.839	8.451	1.00 80.14	BZ00	O
ATOM	3380	CB	ILE	F	108	-31.047	-15.049	5.395	1.00 85.30	BZ00	C
ATOM	3381	CG1	ILE	F	108	-31.210	-15.737	4.041	1.00 87.25	BZ00	C
ATOM	3382	CG2	ILE	F	108	-30.655	-13.589	5.206	1.00 86.34	BZ00	C
ATOM	3383	CD1	ILE	F	108	-32.315	-15.148	3.201	1.00 88.84	BZ00	C
ATOM	3384	N	VAL	F	109	-29.257	-14.424	8.114	1.00 67.56	BZ00	N
ATOM	3385	CA	VAL	F	109	-29.267	-13.831	9.446	1.00 66.53	BZ00	C
ATOM	3386	C	VAL	F	109	-29.155	-12.320	9.293	1.00 65.14	BZ00	C
ATOM	3387	O	VAL	F	109	-28.239	-11.826	8.627	1.00 66.71	BZ00	O
ATOM	3388	CB	VAL	F	109	-28.119	-14.365	10.326	1.00 65.23	BZ00	C
ATOM	3389	CG1	VAL	F	109	-28.172	-13.730	11.705	1.00 65.28	BZ00	C
ATOM	3390	CG2	VAL	F	109	-28.173	-15.882	10.429	1.00 53.85	BZ00	C
ATOM	3391	N	ARG	F	110	-30.079	-11.587	9.909	1.00 66.27	BZ00	N
ATOM	3392	CA	ARG	F	110	-30.068	-10.132	9.872	1.00 64.56	BZ00	C
ATOM	3393	C	ARG	F	110	-30.051	-9.574	11.289	1.00 69.42	BZ00	C
ATOM	3394	O	ARG	F	110	-30.527	-10.211	12.233	1.00 73.34	BZ00	O
ATOM	3395	CB	ARG	F	110	-31.274	-9.587	9.102	1.00 63.19	BZ00	C
ATOM	3396	CG	ARG	F	110	-31.304	-10.010	7.645	1.00 66.27	BZ00	C
ATOM	3397	CD	ARG	F	110	-32.491	-9.403	6.919	1.00 73.55	BZ00	C
ATOM	3398	NE	ARG	F	110	-32.536	-9.784	5.510	1.00 72.91	BZ00	N
ATOM	3399	CZ	ARG	F	110	-33.103	-10.896	5.057	1.00 75.09	BZ00	C
ATOM	3400	NH1	ARG	F	110	-33.670	-11.745	5.903	1.00 86.88	BZ00	N
ATOM	3401	NH2	ARG	F	110	-33.103	-11.163	3.760	1.00 71.91	BZ00	N
ATOM	3402	N	SER	F	111	-29.495	-8.372	11.428	1.00 66.63	BZ00	N
ATOM	3403	CA	SER	F	111	-29.302	-7.731	12.724	1.00 63.36	BZ00	C
ATOM	3404	C	SER	F	111	-29.941	-6.353	12.692	1.00 64.43	BZ00	C
ATOM	3405	O	SER	F	111	-29.555	-5.509	11.878	1.00 59.78	BZ00	O
ATOM	3406	CB	SER	F	111	-27.814	-7.617	13.062	1.00 60.96	BZ00	C
ATOM	3407	OG	SER	F	111	-27.606	-6.686	14.107	1.00 63.33	BZ00	O
ATOM	3408	N	PHE	F	112	-30.896	-6.113	13.584	1.00 72.39	BZ00	N
ATOM	3409	CA	PHE	F	112	-31.619	-4.849	13.622	1.00 76.45	BZ00	C
ATOM	3410	C	PHE	F	112	-31.302	-4.097	14.907	1.00 79.06	BZ00	C
ATOM	3411	O	PHE	F	112	-31.468	-4.635	16.006	1.00 81.06	BZ00	O
ATOM	3412	CB	PHE	F	112	-33.125	-5.079	13.498	1.00 76.97	BZ00	C
ATOM	3413	CG	PHE	F	112	-33.534	-5.738	12.208	1.00 76.22	BZ00	C
ATOM	3414	CD1	PHE	F	112	-33.585	-7.118	12.108	1.00 74.36	BZ00	C
ATOM	3415	CD2	PHE	F	112	-33.873	-4.978	11.099	1.00 75.93	BZ00	C
ATOM	3416	CE1	PHE	F	112	-33.965	-7.730	10.928	1.00 71.63	BZ00	C
ATOM	3417	CE2	PHE	F	112	-34.253	-5.585	9.915	1.00 73.25	BZ00	C
ATOM	3418	CZ	PHE	F	112	-34.299	-6.962	9.831	1.00 70.80	BZ00	C
ATOM	3419	N	SER	F	113	-30.854	-2.853	14.760	1.00 85.08	BZ00	N
ATOM	3420	CA	SER	F	113	-30.512	-2.018	15.898	1.00 83.13	BZ00	C
ATOM	3421	C	SER	F	113	-31.758	-1.349	16.471	1.00 78.62	BZ00	C
ATOM	3422	O	SER	F	113	-32.824	-1.310	15.849	1.00 72.29	BZ00	O
ATOM	3423	CB	SER	F	113	-29.486	-0.956	15.499	1.00 89.41	BZ00	C
ATOM	3424	OG	SER	F	113	-30.017	-0.068	14.527	1.00 97.64	BZ00	O

ATOM	3060	N	VAL	E	411	-28.718	-2.728	6.940	1.00 56.68		N
ATOM	3061	CA	VAL	E	411	-28.938	-3.810	7.894	1.00 54.43		C
ATOM	3062	C	VAL	E	411	-27.811	-4.822	7.761	1.00 53.94		C
ATOM	3063	O	VAL	E	411	-27.601	-5.370	6.669	1.00 79.19		O
ATOM	3064	CB	VAL	E	411	-30.299	-4.485	7.660	1.00 54.31		C
ATOM	3065	CG1	VAL	E	411	-30.557	-5.546	8.717	1.00 65.86		C
ATOM	3066	CG2	VAL	E	411	-31.408	-3.450	7.632	1.00 62.60		C
ATOM	3067	N	PRO	E	412	-27.059	-5.094	8.822	1.00 54.30		N
ATOM	3068	CA	PRO	E	412	-26.050	-6.157	8.744	1.00 64.71		C
ATOM	3069	C	PRO	E	412	-26.680	-7.502	8.421	1.00 63.53		C
ATOM	3070	O	PRO	E	412	-27.462	-8.042	9.210	1.00 66.31		O
ATOM	3071	CB	PRO	E	412	-25.416	-6.146	10.140	1.00 54.66		C
ATOM	3072	CG	PRO	E	412	-25.656	-4.765	10.648	1.00 56.96		C
ATOM	3073	CD	PRO	E	412	-26.991	-4.359	10.095	1.00 55.14		C
ATOM	3074	N	THR	E	413	-26.350	-8.042	7.251	1.00 58.28		N
ATOM	3075	CA	THR	E	413	-26.878	-9.318	6.798	1.00 56.49		C
ATOM	3076	C	THR	E	413	-25.736	-10.307	6.627	1.00 58.40		C
ATOM	3077	O	THR	E	413	-24.662	-9.950	6.136	1.00 60.00		O
ATOM	3078	CB	THR	E	413	-27.631	-9.172	5.473	1.00 52.50		C
ATOM	3079	OG1	THR	E	413	-28.513	-8.044	5.537	1.00 52.22		O
ATOM	3080	CG2	THR	E	413	-28.443	-10.424	5.187	1.00 52.97		C
ATOM	3081	N	LEU	E	414	-25.976	-11.552	7.032	1.00 60.39		N
ATOM	3082	CA	LEU	E	414	-24.992	-12.620	6.922	1.00 63.37		C
ATOM	3083	C	LEU	E	414	-25.583	-13.738	6.079	1.00 65.68		C
ATOM	3084	O	LEU	E	414	-26.578	-14.359	6.473	1.00 57.71		O
ATOM	3085	CB	LEU	E	414	-24.583	-13.150	8.296	1.00 63.54		C
ATOM	3086	CG	LEU	E	414	-23.719	-14.412	8.252	1.00 58.13		C
ATOM	3087	CD1	LEU	E	414	-22.328	-14.085	7.734	1.00 53.54		C
ATOM	3088	CD2	LEU	E	414	-23.647	-15.073	9.618	1.00 58.04		C
ATOM	3089	N	LYS	E	415	-24.973	-13.991	4.927	1.00 69.16		N

ATOM	3090	CA	LYS	E	415	-25.400	-15.050	4.026	1.00	69.93	C
ATOM	3091	C	LYS	E	415	-24.383	-16.180	4.085	1.00	75.55	C
ATOM	3092	O	LYS	E	415	-23.183	-15.953	3.887	1.00	76.33	O
ATOM	3093	CB	LYS	E	415	-25.561	-14.523	2.600	1.00	70.42	C
ATOM	3094	CG	LYS	E	415	-26.889	-14.902	1.969	1.00	73.63	C
ATOM	3095	CD	LYS	E	415	-27.375	-13.840	0.996	1.00	76.05	C
ATOM	3096	CE	LYS	E	415	-28.784	-14.155	0.513	1.00	77.49	C
ATOM	3097	NZ	LYS	E	415	-29.329	-13.102	-0.386	1.00	77.56	N

REVIEWERS' COMMENTS:

Reviewer #1 (Remarks to the Author):

The authors were responsive to the reviewers' comments. The revised manuscript has been improved and thus is recommended for consideration for acceptance.

Reviewer #2 (Remarks to the Author):

The authors have addressed all my comments and concerns, and I am happy to suggest this manuscript for publication.